# SCALING LAW WITH LEARNING RATE ANNEALING

## ABSTRACT

We find that the cross-entropy loss curves of neural language models empirically adhere to a scaling law with learning rate (LR) annealing over training steps:

$$L(s) = L_0 + A \cdot S_1^{-\alpha} - C \cdot S_2,$$

where $L(s)$ is the validation loss at step $s$, $S_1$ is the area under the LR curve, $S_2$ is the LR annealing area, and $L_0$, $A$, $C$, $\alpha$ are constant parameters. This formulation accounts for two main effects: (1) power-law scaling over data size, and (2) the additional loss reduction during LR annealing. Unlike previous studies that only fit losses at final steps, our formulation captures the entire training curve, allowing for parameter fitting using losses from any training step. Applying the scaling law with LR annealing and fitting only one or two training curves, we can accurately predict the loss at any given step under any learning rate scheduler (LRS). This approach significantly reduces computational cost in formulating scaling laws while providing more accuracy and expressiveness. Extensive experiments demonstrate that our findings hold across a range of hyper-parameters and model architectures and can extend to scaling effect of model sizes. Moreover, our formulation provides accurate theoretical insights into empirical results observed in numerous previous studies, particularly those focusing on LR schedule and annealing. We believe that this work is promising to enhance the understanding of LLM training dynamics while democratizing scaling laws, and it is helpful to guide both research and industrial participants in refining training strategies for further LLMs.

## 1 INTRODUCTION

In recent years, large language models (LLMs) have garnered significant academic and industrial attention (Brown et al., 2020; Touvron et al., 2023). The scaling law suggests that the validation loss of language models follow a power-law pattern as model and data sizes increase (Hestness et al., 2017; Kaplan et al., 2020; Henighan et al., 2020). This law provides a powerful framework for forecasting LLM performances before large scale training by fitting losses at smaller scales (OpenAI, 2023; DeepSeek-AI, 2024; Dubey et al., 2024). Numerous studies have explored on the formulation to model the scaling effect of LLMs under various different settings (Bahri et al., 2021; Hernandez et al., 2021; Caballero et al., 2022; Michaud et al., 2023; Muennighoff et al., 2023).

However, typical scaling law formulations focus only on the final loss at the end of training (Hoffmann et al., 2022). Specifically, previous approaches generally rely on a set of training runs and fit the scaling law curve solely on the final loss from each run, while ignoring middle losses during training which do not follow traditional scaling laws. This approach underutilizes the training compute and fails to capture the *training dynamics* within each run. Further, the application of scaling laws in LLM developments is limited since the loss curve through the whole training process is not modeled. An expressive formulation that models full loss curves enables prediction of future training dynamics and also offers insights on understanding the learning process of LLMs.

In this study, we propose a scaling law that models the full loss curve within a complete LLM training run. Specifically, we dive deeper into the training dynamics during LR annealing, and incorporate a LR annealing factor into the traditional scaling law formula to formulate the process. This design is motivated by the observed correlation between LRS and loss curves, where loss gradually decreases as we consume more training steps [1] and then sharply declines when the LR

---

[1]In this paper, we use training steps to quantify the amount of consumed data, as they are typically proportional, with data amount calculated as training steps multiplied by batch size.

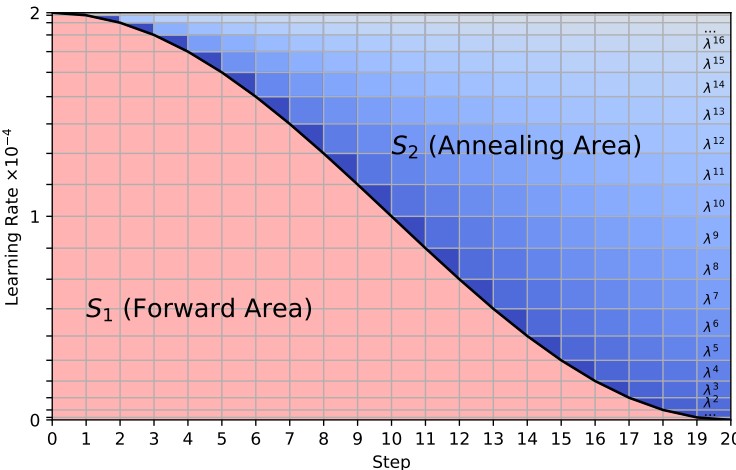

Figure 1: Visualization of $S_1$ and $S_2$ at the 20-th step of a cosine LR scheduler. $S_1$ is the forward area, i.e., sum of red grid areas; $S_2$ is the decayed annealing area, i.e., weighted sum of blue grid areas, where lighter shades indicate smaller weights. Both $S_1$ and $S_2$ contribute to loss reduction, and balancing their values is crucial for achieving the lowest possible final loss.

undergoes significant annealing (Loshchilov & Hutter, 2016; Smith et al., 2018; Ibrahim et al., 2024; Hu et al., 2024). We propose that the model's validation loss $L(s)$ at step $s$ is determined by two main factors: the forward area $S_1$ under the LR curve and the degree of LR annealing $S_2$:

$$L(s) = L_0 + A \cdot S_1^{-\alpha} - C \cdot S_2,$$

$$S_1 = \sum_{i=1}^{s} \eta_i, \qquad S_2 = \sum_{i=1}^{s} \sum_{k=1}^{i} (\eta_{k-1} - \eta_k) \cdot \lambda^{i-k}, \tag{1}$$

where $\eta_i$ is the learning rate at step $i$, and $\lambda$ is a hyper-parameter representing the decay factor for LR annealing momentum (see Sec. 3 in detail),which typically ranges from 0.99 to 0.999. $L_0$, $A$, $C$, $\alpha$ are undetermined positive constants. $S_1$ is also known as the summed learning rate (Kaplan et al., 2020), and $S_2$ represents the LR annealing area. A visualization of $S_1$ and $S_2$ is provided as Fig. 1.

Eq. 1 describes how loss changes for each step in a full loss curve during training. In Eq. 1, the term $L_0 + A \cdot S_1^{-\alpha}$ represents a rectified scaling law that captures the expected loss decreases as a power-law function of the number of training steps. The new term $-C \cdot S_2$ accounts for the further loss drop due to learning rate annealing. Remarkably, this simple formulation accurately describes the validation loss at any training step across various LRS and even allows us to predict the loss curve for unseen LRS. For example, we can fit Eq. 1 to the full loss curve of constant and cosine LRS with 20K total steps (Fig. 2), and then predict the full loss curve for various unseen LRS with longer total steps (e.g. 60K) (Fig. 3).

We validate our proposed equation through extensive experiments and find that: (1) Our formulation performs consistently well across various hyper-parameters and model architectures; (2) Eq. 1 can be extended to incorporate other scaling factors, such as model sizes; (3) Our proposed equation accurately fits the loss curves of open-sourced models; (4) Our formulation can be used to verify and explain numerous previous findings regarding LR annealing and scheduling.

In Sec. 3, we derive the scaling law formulation with LR annealing and elucidate the potential theory underpinning our formulation. Extensive experiments are conducted to validate the formulation. In Sec. 4, we apply our formulation to verify and explain the empirical results from various previous studies. Our approach offers theoretical insights into the crux of loss drop, LR schedule, and LR annealing, enabling LLM participants to better understand training dynamics of LLM and select optimal training recipes in advance. In Sec. 5, we compare our approach to typical scaling law formula, such as the Chinchilla scaling law (Hoffmann et al., 2022). We show that our formulation is more general and requires significantly less compute to fit, which greatly democratizes the development of LLMs and scaling laws.

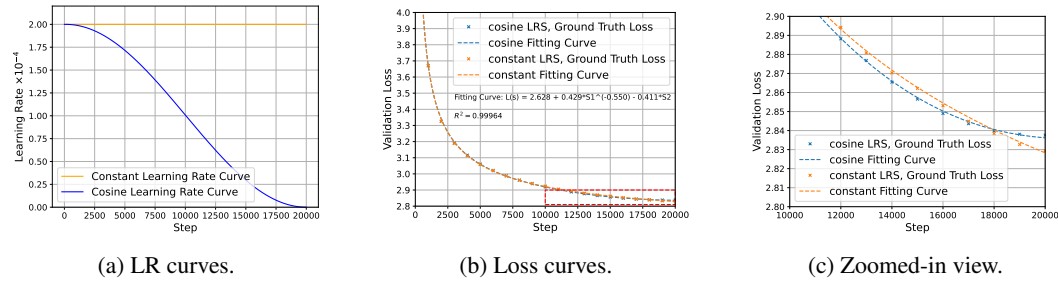

(a) LR curves.  (b) Loss curves.  (c) Zoomed-in view.

Figure 2: Using Eq. 1 to **fit** full loss curves yield by constant and cosine LRS. Total steps = 20K, $\eta_{max} = 2 \times 10^{-4}$, $\eta_{min} = 0$. The fitted equation is $L(s) = 2.628 + 0.429 \cdot S_1^{-0.550} - 0.411 \cdot S_2$.

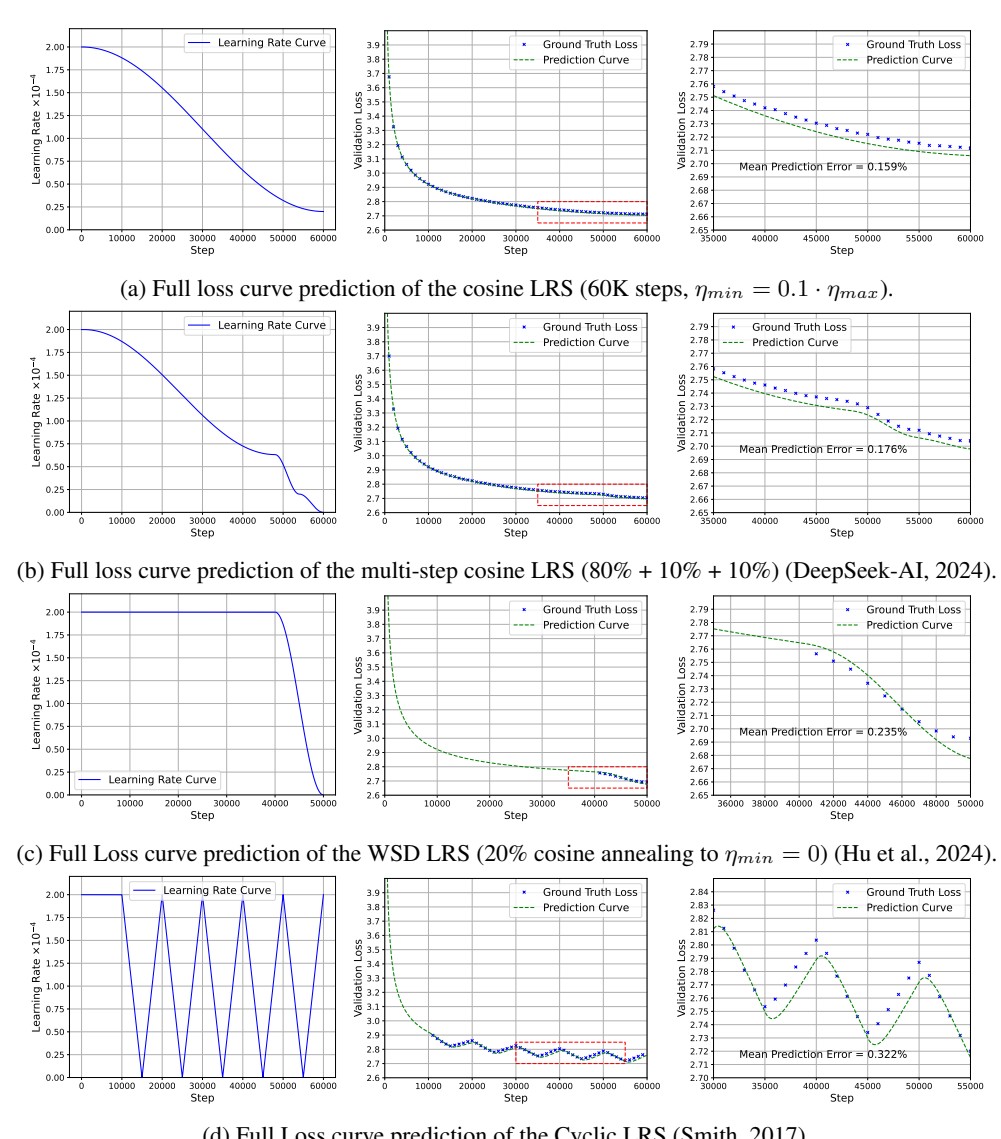

(a) Full loss curve prediction of the cosine LRS (60K steps, $\eta_{min} = 0.1 \cdot \eta_{max}$).

(b) Full loss curve prediction of the multi-step cosine LRS (80% + 10% + 10%) (DeepSeek-AI, 2024).

(c) Full Loss curve prediction of the WSD LRS (20% cosine annealing to $\eta_{min} = 0$) (Hu et al., 2024).

(d) Full Loss curve prediction of the Cyclic LRS (Smith, 2017).

Figure 3: Using the fitted equation from Fig. 2 to **predict** full loss curves for unseen LRS with 60K total steps. The left, middle, and right columns present the LR curve, the loss curve, and a zoomed-in view of loss curve, respectively. Warmup steps (500) are not shown in this figure. The fitted equation accurately predicts each loss curve, particularly for capturing the trend of loss changes as the LR varies. Notable, all LRS and loss curves shown here were **unseen** during the fitting in Fig. 2. The mean prediction errors across different LRS is as low as $\sim 0.2\%$.

## 2 PRELIMINARY

**Scaling Laws.** Cross-entropy loss of language models on the validation set is a reliable indicator of LLMs' performance on downstream tasks (Caballero et al., 2022; Du et al., 2024). Kaplan et al. (2020) empirically discovered a power-law relationship between validation loss $L$ and three factors: model size $N$, dataset size $D$, and training compute. As an application of scaling law, Hoffmann et al. (2022) developed Chinchilla, a compute-optimal LLM, by balancing model size and dataset size. They used a simplified and intuitive scaling law equation: $L(D, N) = L_0 + A \cdot D^{-\alpha} + B \cdot N^{-\beta}$, where $L_0$, $A$, $B$, $\alpha$, $\beta$ are positive constants. Traditional scaling law formulations fit only the loss at the final training step, while ignoring losses from other steps. Collecting a new loss value of data size requires launching a another training run with the same LRS, which is resource-intensive.

**Learning Rate Annealing.** Learning rate annealing is a widely-used technique in training neural networks, where the learning rate is progressively reduced from a maximum to a minimum value following a pre-defined LRS. Various LRS schemes have been proposed to improve the performance and stability of model training (Loshchilov & Hutter, 2016). For example, the popular cosine LRS reduces the LR in a cosine-like pattern over full training steps. WSD LRS (Hu et al., 2024) keeps a constant LR for the majority of training, and applies annealing only in the final (e.g. $10\% \sim 20\%$) steps. In LLM training, it has been widely observed that a more pronounced decrease in the learning rate often results in a more precipitous drop in the validation loss.

## 3 OBSERVATIONS AND EXPERIMENTS

In this section, we elaborate the origin, the intuition, and the experimental basis behind Eq. 1. We then validate our formula through extensive experiments.

### 3.1 SIMILARITY BETWEEN LEARNING RATE, GRADIENT NORM, AND LOSS

The first key observation is that the shapes of LR curve, gradient norm curve, and validation loss curve are quite similar across various LRS when training LLMs (Fig. 4). This suggests an implicit connection between learning rate and loss, where gradient norm could be the bridge.

**Scaling Laws for Constant LRS.** A constant LRS is a special LRS, where every training step can be viewed as an endpoint of the LRS. Notably, the Chinchilla scaling law (Hoffmann et al., 2022) exactly fits losses of last steps, i.e., LRS endpoints. Therefore, it is expected that the validation loss of all steps under a constant LRS adheres to the Chinchilla scaling law, i.e., a power-law over training step $s$.

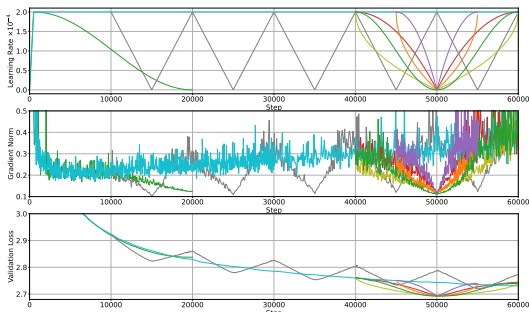

Figure 4: The shapes of LR (top), gradient norm (medium), and validation loss (bottom) curves exhibit high similarity across various LRS (labeled as different colors).

**Extra Loss Changes in LR Annealing.** Unlike a constant LRS, LR annealing (or re-warmup) brings significant local changes in the loss (see Fig. 4), causing the full loss curve to deviate from the traditional power-law formulation that consider only the training steps $s$. We hypothesis that such loss changes can be captured by an additional LR ($\eta$) related term, i.e.,

$$L(s) = L_0 + A \cdot s^{-\alpha} - f(\eta), \tag{2}$$

where the first two terms follow traditional scaling laws, while the last term denotes the extra loss change brought by LR annealing. Recall the similarity between learning rate and loss curves, we can form an initial guess for $f(\eta)$ as $f(\eta) = C \cdot \eta$, where $C$ is a positive constant.

**Training Discount in Annealing.** The form of Eq. 2 is still imperfect. Note that the gradient norm $\|\mathbf{g}\|$ decreases almost proportionally with LR during the annealing process (shown in Fig. 4). Thus, the amount of parameter movement (approximately $\eta \cdot \|\mathbf{g}\|$ per step) in the LR annealing stage declines at an almost quadratic rate compared to stages before annealing. As the parameter movement become smaller, the change in loss also slows down accordingly. Therefore, the loss drop

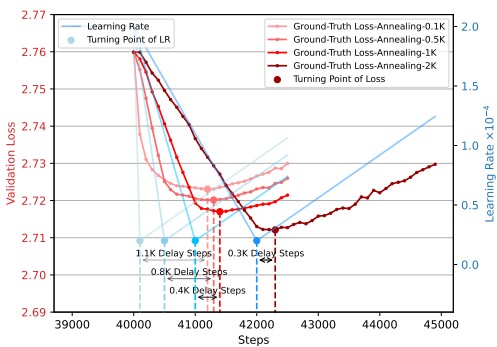 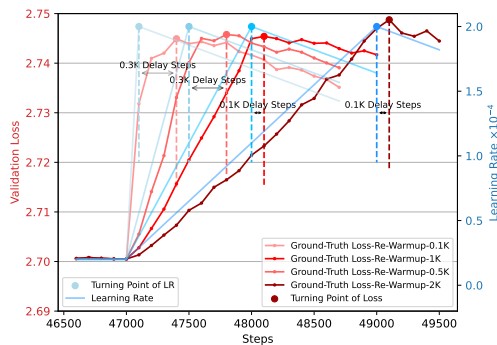

(a) Different delay steps in the annealing process associated with different annealing steps (0.1K, 0.5K, 1K and 2K).

(b) Different delay steps in the re-warmup process associated with different re-warmup steps (0.1K, 0.5K, 1K and 2K).

Figure 5: The delay phenomenon between the LR and validation loss curves. This phenomenon suggests that LR annealing (re-warmup) has momentum.

brought by the power law term (i.e., the first two terms in Eq. 2) should also diminish during LR annealing. This consideration leads to an improved equation:

$$L(s) = L_0 + A \cdot S_1^{-\alpha} - f(\eta), \qquad S_1 = \sum_{i=1}^{s} \eta_i, \tag{3}$$

where $S_1$ is the forward area, i.e., the area under the LR curve (as visualized in Fig.1), which could be approximately interpreted as the total amount of parameter updates.

## 3.2 LR ANNEALING MOMENTUM

Another key observation is that LR annealing has momentum. To refine the formulation of $f(\eta)$, we design a special LRS where the LR decreases linearly from $\eta_{max}$ to $\eta_{min}$ and then increases. The increasing stage always has a fixed slope, reaching the maximum value in 5K steps, while the slope of the decreasing stage is varied, with durations of 0.1K, 0.5K, 1K, and 2K. Symmetrically, we design another LRS where the LR increases linearly from $\eta_{min}$ to $\eta_{max}$ and then decreases. Fig. 5 shows the corresponding LR and loss curves.

We observe a **delay phenomenon** between the LR and the validation loss. Firstly, the turning point of the validation loss curve consistently lags behind the turning point of the LR curve, indicating that the validation loss continuous along its previous trajectory for some steps even after the LR changes direction. Secondly, the steeper the slope of the LR annealing (or re-warmup), the more pronounced the delay phenomenon becomes. Thirdly, given the same LR slope, the left figure (where LR decreases then increases) consistently shows a longer delay compared to the right figure (where LR increases then decreases).

Interestingly, this phenomenon closely resembles the physical experiment of a small ball rolling down a slope. The steeper the slope, the faster the ball accelerates. When the ball lands, the accumulated momentum causes the ball to slide further. Inspired by this delay phenomenon, we hypothesize that $f(\eta)$, the loss reduction induced by LR annealing, has cumulative historical formation so that the past change of learning rate will affect the following loss curve for a few steps. In summary, *learning rate annealing exhibits momentum*. To capture this, we define $f(\eta) = C \cdot S_2$, where $S_2$ is calculated as:

$$m_i = \lambda \cdot m_{i-1} + (\eta_{i-1} - \eta_i),$$
$$S_2 = \sum_{i=1}^{s} m_i = \sum_{i=1}^{s} \sum_{k=1}^{i} (\eta_{k-1} - \eta_k) \cdot \lambda^{i-k}, \tag{4}$$

where $m_i$ is the LR annealing momentum at step $i$ ($m_1 = 0$), and $\Delta\eta = \eta_{i-1} - \eta_i$ denotes the LR annealing amount at step $i$. $\lambda$ is the decay factor that signifies how much historical information is retained. We find that $\lambda$ values between 0.99 and 0.999 generally works well. In contrast, $\lambda = 0$

implies no momentum effect, reducing $f(\eta)$ to $C \cdot \eta_s$, which degenerate to the initial form mentioned above. Note that $S_2$ applies not only to LR annealing ($S_2 > 0$), but also to LR re-warmup ($S_2 < 0$). This means that our equation is applicable to scenarios like continual pre-training, where LR re-warmup plays an important role in improving outcomes. Fig. 1 presents a visualization of $S_2$.

### 3.3 FINAL FORMULATION

We formally present our formulation for the scaling law with LR annealing:

**Scaling Law with LR Annealing.** *Given the same training and validation dataset, the same model size, the same training hyper-parameters such as warmup steps, **max learning rate** $\eta_{max}$ and batch size, the language modeling loss at training step $s$ empirically follows the equation $L(s) = L_0 + A \cdot S_1^{-\alpha} - C \cdot S_2$, where $S_1$ and $S_2$ are defined in Eq. 1. $L_0$, $A$, $C$, $\alpha$ are positive constants.*

Our formulation describes the loss of each training step across different LRS. It allows fitting based on a simpler LRS with shorter training steps and enables the prediction of validation losses for more complex LRS with longer training steps. Notably, loss curves with different max learning rates have different values of $L_0$, $A$, $C$, $\alpha$, and our scaling law does not fit divergent and collapsed loss curves (e.g., overly large LR). We also discuss some possible corner cases (i.e., $\eta = 0$) in Appendix H.3.

**Loss Surface as a Slide.** To better understand our formulation, we view the loss surface of language models as a slide in Fig. 6. The optimization process can be seen as sliding down the slide according to the power-law scaling (orange line), while oscillating on the inner wall (blue dashed line). When the LR anneals (red line), the amplitude of the oscillation decreases, resulting in a reduction in loss.

**Balance between $S_1$ and $S_2$.** Note that in Eq.1, $\frac{\partial L}{\partial S_1} < 0$ and $\frac{\partial L}{\partial S_2} < 0$ always hold, indicating that increases in both $S_1$ and $S_2$ help to reduce the loss. However, as shown intuitively in Fig. 1, there exists delicate balance between $S_1$ and $S_2$. When LR begins to anneal and $S_2$ starts to increase, the forward area $S_1$ of subsequent steps starts to diminish instead. Our equation aptly describes this delicate balance. In Sec. 4, we elaborate this topic in detail.

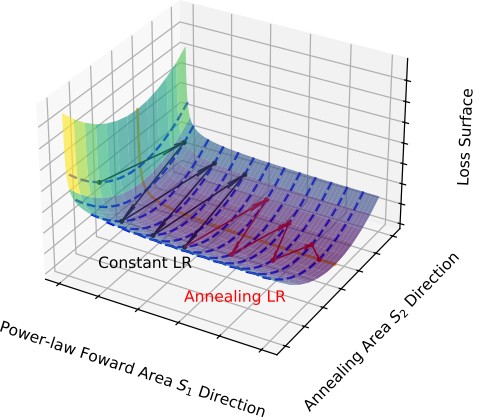

Figure 6: Loss surface of language models as a *slide* after simplification. Optimization direction could be decomposed into two directions: power-law scaling direction ($S_1$, sliding down) and annealing direction ($S_2$, inner height of the slide).

### 3.4 EXPERIMENTS

**LR Warmup.** LR warmup is important for training LLMs. During the warmup stage, neural networks are prone to random optimization, resulting in *unpredictable* outcomes (Hestness et al., 2017). Various studies, along with our own pilot experiments (Appendix A), show that LR warmup significantly accelerates model convergence. High gradient norms are usually observed during the LR warmup stage, especially in the initial steps of training (see Fig. 4). This indicates that model parameters undergo substantial updates during this stage. Therefore, in all our experiments, we linearly warmup LR to reach $\eta_{max}$ and compute $S_1$ and $S_2$ assuming a constant LR value $\eta_{max}$ in the warmup stage.

**Experimental Setups and Fitting Details.** We use standard experimental setups for LLM pretraining. To verify the robustness of our formulation across different experimental settings, we have five distinct experimental setups (see Appendix C). We adopt $\lambda = 0.999$ in our all experiments. We follow the fitting approach of Hoffmann et al. (2022) to obtain parameters in our equation (see Appendix B for more details).

**Fitting and Prediction Results.** We fit Eq.1 on the loss curves under constant and cosine LRS with 20K total steps (see Fig. 2), and then predict the full loss curves under several unseen LRS with 60K total steps (see Fig. 3). The results show an almost perfect fit, achieving a coefficient of determination ($R^2$) greater than 0.999. This underscores the robust capability of our equation to accurately fit loss curves across diverse LRS using a single parameter tuple.

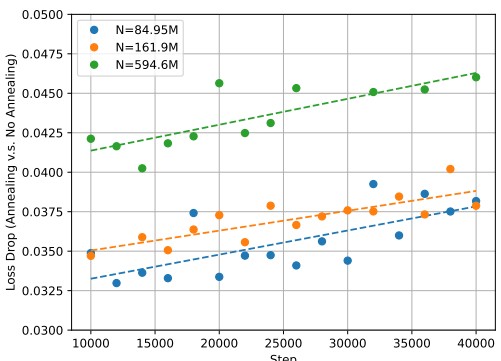 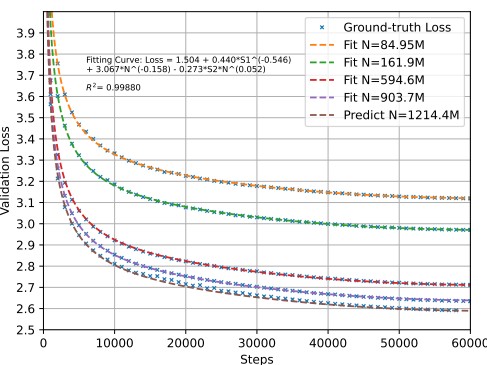

(a) The loss drop brought by LR annealing for different model size $N$. Dashed lines represent trends over steps. The loss drop brought by LR annealing scales with data size and model size.

(b) Curve fitting and prediction on cosine LRS (60K steps to $\eta_{min} = 0.1 \cdot \eta_{max}$) of different model sizes using our $N$-extended scaling law. Results for N=1214.4M are predicted.

Figure 7: The loss drop brought by LR annealing (left) and the $N$-extended full loss curve fitting and prediction (right).

The prediction results in Fig. 3 indicate that our formulation is broadly applicable and generalizes robustly across four unseen LRS, with a mean prediction error as low as 0.2%. Moreover, our equation can accurately predict losses even for complex LRS that include multiple LR re-warmup stages (Fig. 3d), despite that the loss curves used for fitting do not contain any LR re-warmup stages.

**Extensive Experiments on Different Setups.** To demonstrate the broad applicability of our proposed equation, we conduct additional fitting and prediction experiments using various setups. (1) We use an alternative set of training hyper-parameters (Appendix D.1); (2) We test our equation on the Mixture of Experts (MoE) architecture (Appendix D.2); (3) We apply our equation to predict loss curves for a much longer training run involving a 1.7B parameter model trained on 1.4T tokens (Appendix D.3). (4) We fit the loss curves of open-sourced models, including BLOOM-176B trained on 300B tokens (BigScience, 2022) and OLMo-1B trained on 2T tokens (Groeneveld et al., 2024) (Appendix D.4). All experiments produce excellent results, indicating that our equation is effective across diverse experimental setups, including different training hyper-parameters, architecture, model sizes, and dataset scales. We also present the ablation studies on $S_1$ and $S_2$ in Appendix D.5, which shows each component in our formulation is important and indispensable.

3.5 EXTENSION TO MODEL SIZE SCALING

**Loss Drop During Annealing Scales with Model Size $N$.** We explore the effect of model size $N$ on the loss drop during the annealing stage. Specifically, we compare the final losses obtained with a constant LRS and a WSD LRS (10% cosine annealing to $\eta_{min} = 0$) to estimate the loss drop due to LR annealing. We conduct this experiment on different total steps and different model sizes. The experimental results are shown in Fig. 7a. It suggest that the loss drop from LR annealing scales with both annealing steps and model sizes. This implies that the annealing area $S_2$ in our equation should also increase as the model size $N$ increases. We suppose there is a simple relationship of $S_2 \propto N^\gamma$ where $\gamma$ is a positive constant.

**Model Size Scaling.** Building on the experiments and analysis above, we extend our proposed Eq.1 to incorporate model size scaling, based on traditional scaling laws:

$$L(s, N) = L_0 + A \cdot S_1^{-\alpha} + B \cdot N^{-\beta} - C \cdot S_2 \cdot N^\gamma, \tag{5}$$

where $N$ is the number of non-embedding model parameters, and $B$, $\beta$, $\gamma$ are positive constants related to $N$. We realize $S_2 \propto N^\gamma$ via a multiplier $N^\gamma$ to the original annealing term $-C \cdot S_2$.

**Fitting and Prediction with Model Size.** We validate Eq. 5 by fitting the full loss curves of models with varying sizes. We then apply the obtained equation to predict full loss curve on the unseen largest model size. Results in Fig. 7b show an almost perfect fit ($R^2 > 0.998$) and prediction for entire training dynamics of larger-scale models. This indicates the effectiveness and robustness of our proposed $N$-extended equation. Additional $N$-extended experiments with other setups further confirm the robustness of our formulation (see detail in Appendix D.6).

## 4 APPLICATION

We apply our proposed formulation to validate and provide a theoretical explanation for numerous existing experimental findings regarding the training dynamics of language models. These key insights also guide researchers in selecting critical LRS before initiating model training. An interesting summary is that, *the art of learning rate schedule lies in the delicate balancing act between forward area and annealing area.*

### 4.1 DETERMINING COSINE CYCLE LENGTH AND MINIMUM LR IN COSINE LRS.

Many papers have found that in LLM pre-training using cosine LRS, setting the cosine cycle length $T$ as the total steps $S$, and setting min LR as nearly 0 (rather than 10% max LR) can lead to the optimal loss (Hoffmann et al., 2022; Hu et al., 2024; Hägele et al., 2024; Parmar et al., 2024). We theoretically validate this observation using our equation in Fig. 8. The predicted loss curve with $T = S$ and a minimum LR of 0 indeed achieves the optimal loss in the final step. Moreover, our equation gives a quite intuitive explanation: setting $T > S$ leads to incomplete annealing, while $T < S$ leads to a small forward area $S_1$ due to early annealing. Thus, the optimal configuration is to set $T$ equal to $S$. Also, setting the minimum LR to 0 maximizes the annealing amount, thereby increasing the annealing area $S_2$, which facilitates lower final loss.

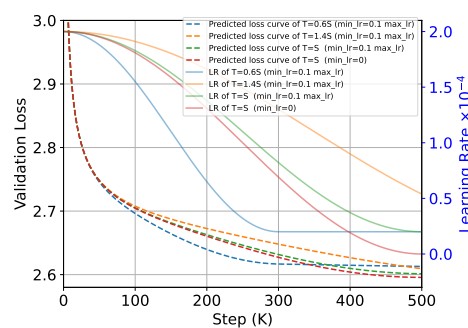

Figure 8: Predicted loss curves of different cycle length $T$ and min LR in cosine LRS. The results well align with previous studies.

### 4.2 IT VERIFIES AND EXPLAINS WHY WSD AND MULTI-STEP COSINE LRS ARE BETTER.

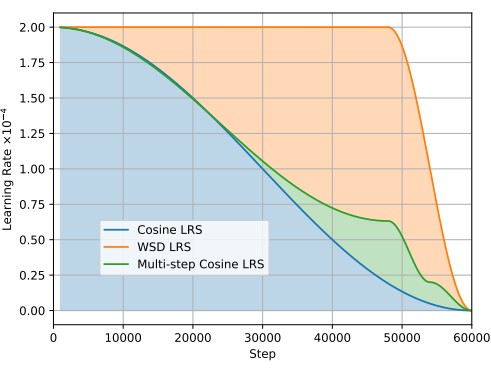

(a) Learning rate curves of three types of LRS.

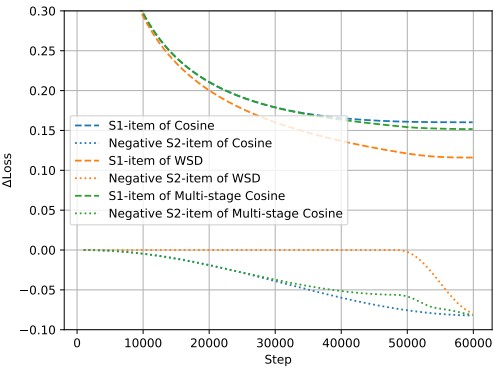

(b) $S_1$-item and negative $S_2$-item of different LRS.

Figure 9: The comparison between $S_1$-item and negative $S_2$-item in different LRS.

Recent studies have shown that WSD LRS (Hu et al., 2024) and multi-step cosine LRS (DeepSeek-AI, 2024) result in lower loss compared to the traditional cosine LRS. We validate and elucidate this finding using our proposed equation. Fig. 9 shows the learning rate curve (left) and the predicted loss drop (right) for different LRS. The results suggest that for WSD and multi-step cosine LRS, the negative $S_2$-item ($-C \cdot S_2$) is slightly larger than that of the cosine LRS, whereas the $S_1$-item ($A \cdot S_1^{-\alpha}$) is significantly lower. Specifically, both the WSD LRS and multi-step cosine LRS unintentionally employ strategies that marginally reduces $S_2$ but substantially increases $S_1$, leading to an overall decrease in validation loss.

### 4.3 DETERMINING OPTIMAL ANNEALING RATIO OF WSD SCHEDULER.

In the case of WSD LRS, it is crucial to ascertain the optimal annealing ratio for training steps. Hägele et al. (2024) found that there is an optimal annealing ratio for WSD LRS, and both exces-

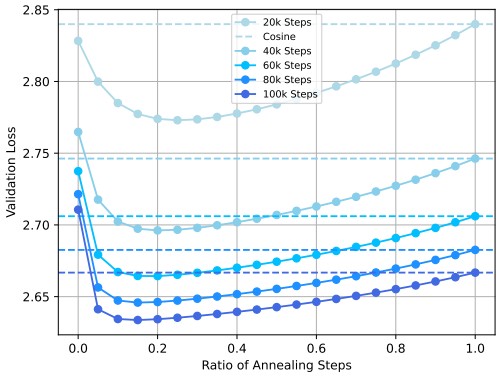 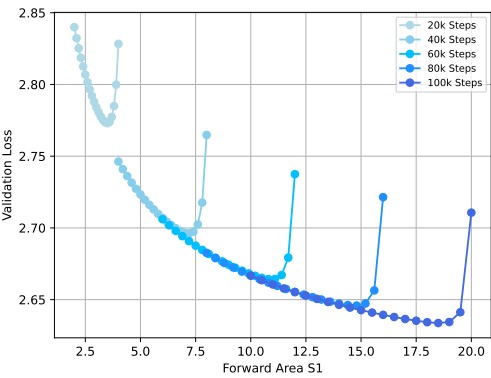

(a) The relationship between the predicted final loss and the ratio of annealing steps under the condition of different total steps.

(b) The relationship between predicted final loss and the forward area $S_1$ of different total steps. Different points denote different annealing ratios.

Figure 10: Illustration of the predicted loss in relation to the ratio of annealing steps and the forward area in WSD LRS (cosine annealing), presenting parabola-like curves, with a distinct optimal loss.

sively high or low annealing ratios lead to sub-optimal model performance. This phenomenon can be further elucidated through our proposed equation. Specifically, a high annealing ratio results in a significant reduction of the forward area $S_1$ while a low annealing ratio leads a diminished annealing area $S_2$. Our scaling law equation describes the trade-off between the forward area $S_1$ and the annealing area $S_2$ about the annealing ratio.

Fig. 10 depicts the final loss predicted by our equation for various annealing ratios and total training steps. The predictions form parabola-like curves, and align well with the actual experimental results reported in previous studies. This suggests that a moderate annealing ratio, typically around 10% to 20%, is optimal, as it balances $S_1$ and $S_2$ to maximize their combined effect, thereby minimizing the overall validation loss. Moreover, our equation can directly predict the optimal annealing ratio for different total steps without large-scale experiments, which saves a lot of resources.

### 4.4 MANY OTHER TAKEAWAYS

Moreover, we use our equation to verify and explain more phenomena as follows: (1) Appendix G.1: An empirical reason of loss dropping more sharply when LR anneals (Loshchilov & Hutter, 2016; Ibrahim et al., 2024; DeepSeek-AI, 2024). (2) Appendix G.2: the comparison between constant and cosine LRS, aligned with previous works (Hu et al., 2024). (3) Appendix G.3: how to choose the optimal annealing function in WSD LRS, aligned with previous works (Hägele et al., 2024). (4) Appendix G.4 and G.5: how to re-warmup (including re-warmup peak LR and steps) in continual pre-training, aligned with previous works (Gupta et al., 2023). Given the instances above, we believe that our equation can help analyze and select more training recipes in specific scenarios.

## 5 COMPARISON WITH CHINCHILLA SCALING LAW

### 5.1 REDUCTION TO CHINCHILLA SCALING LAW

Our scaling law equation can predict the full loss curve across any given LRS. In this section, we show that our equation has no contradiction with traditional scaling laws, and it is a generalized form of the Chinchilla scaling law (Hoffmann et al., 2022). Specifically, all the final loss values for different total training steps following our equation should also follow a power-law relationship. We prove this by dividing two conditions: (1) constant LRS, and (2) other LRS.

**Constant LRS.** In the case of a constant LRS, the annealing area $S_2$ is always zero and the forward area $S_1 = \eta_{max} \cdot s$, where $s$ is the step, and $\eta_{max}$ is the constant maximal LR. Thus, the whole train loss curve becomes: $L(s) = L_0 + (A \cdot \eta_{max}^{-\alpha}) \cdot s^{-\alpha} = L_0 + A' \cdot s^{-\alpha}$, which aligns with the Chinchilla scaling law equation.

**Other LRS.** For non-constant LRS, we use a statistical approach to show that our equation can be reduced to the Chinchilla scaling law. Specifically, we verify whether the Chinchilla scaling law ad-

Table 2: The comparison of computational cost for fitting different scaling law equations.

| Equation | LRS | Computational cost | Applicable to other LRS? |
|---|---|---|---|
| Chinchilla | Cosine | 100% | No |
| Chinchilla | WSD (20% annealing) | 21.6% | No |
| Chinchilla | WSD (10% annealing) | 11.8% | No |
| Ours | Any (except constant) | <1.0% | Yes |

equately fits the endpoints of loss curves predicted by our equation. The parameter tuple of our equation is $(L_0, A, C, \alpha)$. We randomly sample different parameter tuples (detailed in Appendix E.1). Each parameter tuple represents a synthetic fitting result corresponding to a distinct set of experimental setups (e.g., dataset, model size, etc.). For each sampled parameter tuple, we apply our equation to predict the final loss for different total training steps with both cosine and WSD LRS, and then employ the predicted losses to fit the Chinchilla scaling law. We calculate the mean and standard deviation of $R^2$ values for each fit. The results in Table 1 demonstrate that Chinchilla scaling law fits well on the data predicted by our scaling law equation. Thus,

Table 1: Mean and std of $R^2$ for different parameter fits.

| LRS | mean($R^2$) ↑ | std($R^2$) ↓ |
|---|---|---|
| Cosine | 0.972 | 0.056 |
| WSD | 0.979 | 0.053 |

our equation can be considered a generalization that can be reduced to the Chinchilla scaling law.

## 5.2 SCALING LAW FITTING AND PREDICTION DEMOCRATIZATION

Our scaling law equation allows us to utilize all loss values from a full loss curve during training, while traditional scaling laws can only collect a single data point from the full loss curve. This feature allows us to fit scaling laws with much less cost. For a direct comparison, we compare the computational efficiency of our approach and the Chinchilla scaling law (Hoffmann et al., 2022). Specifically, we assume to collect 100 data points for parameter fitting, and estimate the computational costs needed to fit the respective scaling law equations under different LRS configurations (see Table 2). More details can be found in Appendix E.2. The results indicate that our proposed equation uses less than 1% of the computational cost required by the Chinchilla scaling law. Further, our scaling law with LR annealing, can be universally applied to predict loss curves for unseen LRS, thus conserving even more computational resources. This approach significantly democratizes the study of scaling laws in LLM pre-training, paving the way for a more environmentally friendly and carbon-efficient methodology.

## 6 DISCUSSION

(1) We analyze the impact of the decay factor $\lambda$ of our equation in Appendix H.1, and it suggests that selecting a proper decay factor is important for determining the balance point between $S_1$ and $S_2$; (2) We analyze the root reasons of the delay phenomenon mentioned in Sec. 3 in Appendix H.2. It suggests that neither the Adam optimizer (Kingma & Ba, 2015) nor $S_1$ are the root reasons and this can be an important future work; (3) We discuss some potential variation of our proposed equation (e.g. $\eta = 0$ case and $L \propto S_2^\zeta$ variant), and investigate other possible scaling law formats with LR annealing in Appendix H.3. The results validate the superiority of our proposed formula.

## 7 CONCLUSION

In conclusion, we propose that the loss curves of neural language models empirically adhere to a scaling law with learning rate annealing over training steps $s$: $L(s) = L_0 + A \cdot S_1^{-\alpha} - C \cdot S_2$. This equation can accurately predict full loss curves across unseen learning rate schedulers. We present the underlying intuition and theory for deriving our equation and demonstrate that our approach can be extended to capture the scaling effect of model sizes. Extensive experiments demonstrate that our proposed scaling law has good accuracy, scalability, and holds under various experimental setups. It also offers accurate theoretical insights to the training dynamics of LLMs, and explains numerous phenomena observed in previous studies. We believe that the scaling law with LR annealing is promising to reshape the understanding of researchers for LLM training and scaling laws.

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

## A    IMPACT OF WARMUP STEPS

We conduct experiments on the impact of learning rate warmup steps. As shown in Fig. 11, we find that 500 warmup steps can speed up convergence, and get the lowest validation loss compared to 100 or no LR warmup. The finding is aligned with previous works Liu et al. (2020); Kosson et al. (2024). The experimental results also guide us to choose 500 warmup steps in the main experiments of this work [2].

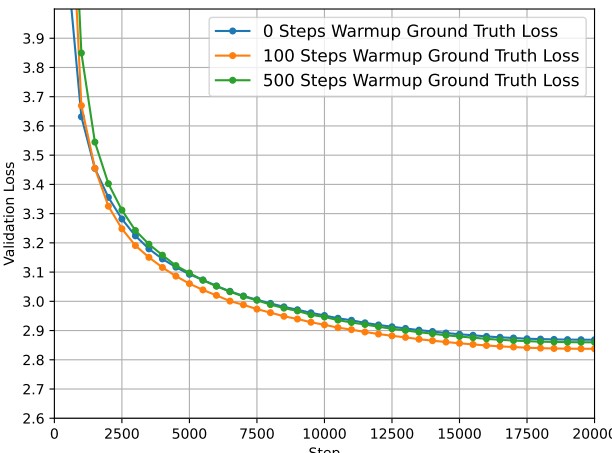

Figure 11: The comparison of the true loss curve of different warmup steps. We experiment on cosine LRS with 20K total steps.

## B    FITTING DETAILS

Given a learning rate scheduler, we can easily compute out $S_1$ and $S_2$ of each step in advance. To estimate $(L_0, A, C, \alpha)$, we adopt a similar fitting method as Chinchilla scaling law (Hoffmann et al., 2022). Specifically, we minimize the Huber loss (Huber, 1964) between the predicted and the observed log loss using the L-BFGS algorithm (Nocedal, 1980):

$$\min_{L_0, A, C, \alpha} \sum_{\text{Step } i} \text{Huber}_\delta \left( \log \hat{L}(i) - \log L(i) \right) \tag{6}$$

We implement this by the utilization of `minimize` in `scipy` library. Huber loss is to enhance to robustness of the fitting results and we set $\delta$ of Huber loss as $1.0 \times 10^{-3}$. We mitigate the potential issue of local minima of fitting by choosing the optimal fit from a range of initial conditions. Note that in practice, we can also fit the full loss curves using multiple LRS with a single tuple of $(L_0, A, C, \alpha)$. In this situation, we sum the Huber losses in Eq. 6 of all fitted LRS.

## C    EXPERIMENTAL SETUPS

In this work, we use multiple sets of experimental setups, in order to validate that our equation can work across different experimental setups. For clarification, we present the experimental setup list as shown in Table 3. In our most experiments, we use the main setting A. Other than the five settings, we also successfully fit our equation on BLOOM's and OLMo's loss curves, and their experimental settings are totally different. Refer to their papers for the experimental settings (BigScience, 2022; Groeneveld et al., 2024).

---

[2]Note LR warmup in training from scratch is different from LR re-warmup in continual training, where we do not regard re-warmup steps as a hyper-parameter and will show how to apply our equation to find optimal re-warmup recipes in Appenidx. G.4 and G.5.

Table 3: Experimental settings adopted in this work. Model size denotes the number of non-embedding paramters. Our datasets include Fineweb (Penedo et al., 2024) and RedPajama-CC (Computer, 2023). * denotes pre-training multilingual dataset including mixture of sources such as common crawls, books, arxiv, code, etc. We use AdamW Optimizer (Kingma & Ba, 2015; Loshchilov & Hutter, 2017), denoted as AO. Most experiments adopt Llama-3's tokenizer (Dubey et al., 2024). Ext Llama-2's is extended from Llama-2's tokenizer (Touvron et al., 2023) by adding vocabulary.

| Setups | Setting A (mainly) | Setting B | Setting C |
|---|---|---|---|
| **Model Size** | 594M | 293M | multiple |
| **Train Dataset** | Fineweb | Finweb | Mixture-train* |
| **Val Dataset** | RedPajama-CC | RedPajama-CC | Mixture-valid* |
| **Total Steps** | 60K | 120K | 143K |
| **Maximal LR** | $2 \times 10^{-4}$ | $2 \times 10^{-4}$ | $1.381 \times 10^{-3}$ |
| **Warmup Steps** | 500 | 100 | 500 |
| **Batch Size (tokens)** | 4M | 2M | 4M |
| **Sequence Length** | 4096 | 4096 | 4096 |
| **Tokenizer** | Llama-3's | Llama-3's | Ext Llama-2's |
| $\beta_1,\beta_2$ **in AO** | 0.9, 0.95 | 0.9, 0.95 | 0.9, 0.95 |
| **Weight Decay** | 0.1 | 0.1 | 0.1 |
| **Gradient Clip** | 1.0 | 1.0 | 1.0 |

| Setups | Setting D (MoE) | Setting E (1.4T tokens) |
|---|---|---|
| **Model Size** | $8 \times 106$M | 1704M |
| **Train Dataset** | Fineweb | Mixture-train* |
| **Val Dataset** | RedPajama-CC | Mixture-valid* |
| **Total Steps** | 60K | 350K |
| **Maximal LR** | $2 \times 10^{-4}$ | $6 \times 10^{-4}$ |
| **Warmup Steps** | 500 | 1000 |
| **Batch Size (tokens)** | 4M | 4M |
| **Sequence Length** | 4096 | 8192 |
| **Tokenizer** | Llama-3's | Llama-3's |
| $\beta_1,\beta_2$ **in AO** | 0.9, 0.95 | 0.9, 0.95 |
| **Top-$k$ Experts** | 2 | - |
| **Auxiliary Loss** | 0.01 | - |
| **Weight Decay** | 0.1 | 0.1 |
| **Gradient Clip** | 1.0 | 1.0 |

## D  OUR SCALING LAW ON EXTENSIVE EXPERIMENTS SETUPS

### D.1  ANOTHER SET OF TRAINING HYPER-PARAMETERS

Fig. 2 and Fig. 3 show that our equation can work very well on our main experimental setup. For proving that our scaling law with LR annealing can apply to different (but given) experimental settings, we change the setting from $A$ to $B$ (refer to Table 3) and observe whether our equation can still work or not. The fitting results are shown in Fig. 12. The prediction results are shown in Fig. 13. The results suggest that our scaling law with LR annealing can still work well across different experimental setups.

### D.2  EXPERIMENTS ON ANOTHER ARCHITECTURE: MOE

Fig. 2 and Fig. 3 show that our equation can work very well on the dense Llama-like architecture (Vaswani et al., 2017; Touvron et al., 2023). We prove that our scaling law can also apply to different model architectures and we replace Dense model with Mixture of Experts (MoE) architecture. We add widely-used auxiliary loss to do load balancing among experts (Fedus et al., 2021). The experimental setting is shown as Setting $D$ in Table 3. Moreover, we change the LRS and total steps to 60K WSD with 10K annealing steps in fitting, testing whether our scaling law is effective under various circumstances. The fitting results are shown in Fig. 14 while the prediction results are shown in Fig. 15. The results suggest that our scaling law can with LR annealing can still work well on MoE architecture.

### D.3  SCALING UP: PREDICTION FOR MUCH LONGER STEPS

Our equation has proven its utility in predicting the validation loss over a significantly large number of total steps. This scalability feature is particularly useful in handling large-scale training scenarios.

To illustrate its effectiveness, we apply our equation to predict the loss curve during the annealing stage of the training process. The model we train is a sizable 1.7 billion parameter model, and the training involved a tremendous number of 1,400 billion total training tokens. This is a considerable scale that tests the practicality and effectiveness of our equation. The specific experimental setup is Setting $E$, which can be found in Table 3.

The fitting and prediction results are shown in Figure 16 and Figure 17 respectively. It shows that we successfully get to know the loss curve in the critical annealing stages after 10x longer steps in advance, which is crucial to handle the relationship between training dynamics and training recipes. For example, Llama-3 adopts annealing to do pre-training data selection (Dubey et al., 2024).

### D.4  OPEN-SOURCED FULL LOSS CURVES

For further verification for our proposed scaling law, we apply our equation on open-sourced language models and the corresponding full loss curves, including BLOOM-176B (BigScience, 2022) and OLMo-1B (Groeneveld et al., 2024). As shown in Fig. 18, our equation also fits very well on the open-sourced model training curves, even when the model size scales up to 176B (e.g. BLOOM) and token number scales up to 2000B over 740K steps (e.g. OLMo).

### D.5  ABLATION STUDIES ON $S_1$ AND $S_2$

In Sec. 3, we present the strong capability of our proposed scaling law. The formulation of our scaling, $L(s) = L_0 + A \cdot S_1^{-\alpha} - C \cdot S_2$, contains two key components including $S_1$ and $S_2$. In this section, we conduct ablation studies on $S_1$ and $S_2$. Specifically, we compare the forms without $S_1$ or $S_2$ using setting A. For each format, we re-fit the full loss curves under 20K cosine + constant and re-predict the full curves on longer steps under different LRS. The prediction error results are shown as Table 4. The results indicate that the prediction error increases significantly in the absence of either $S_1$ or $S_2$ and suggest that each component in our scaling law is important and indispensable.

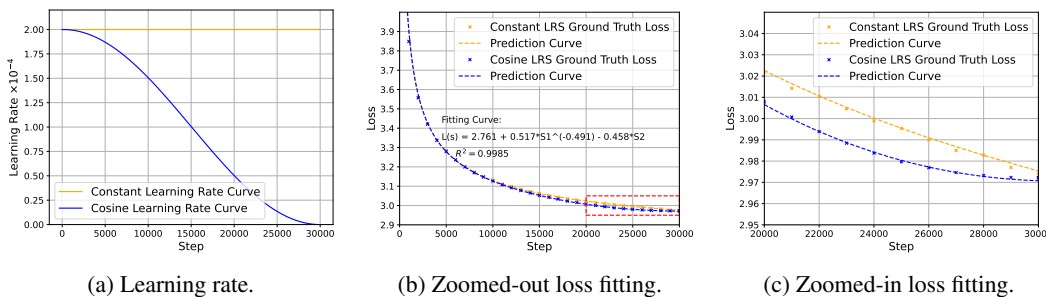

(a) Learning rate.   (b) Zoomed-out loss fitting.   (c) Zoomed-in loss fitting.

Figure 12: Full loss curve **fitting** on cosine (30K steps to $\eta_{min} = 0$) and constant LRS. The figures omit the warmup in the first 100 steps. After fitting, we get a **universal** loss equation $L = 2.761 + 0.517 \cdot S_1^{-0.491} - 0.458 \cdot S_2$. Refer to setting B in Table 3 for experimental setups.

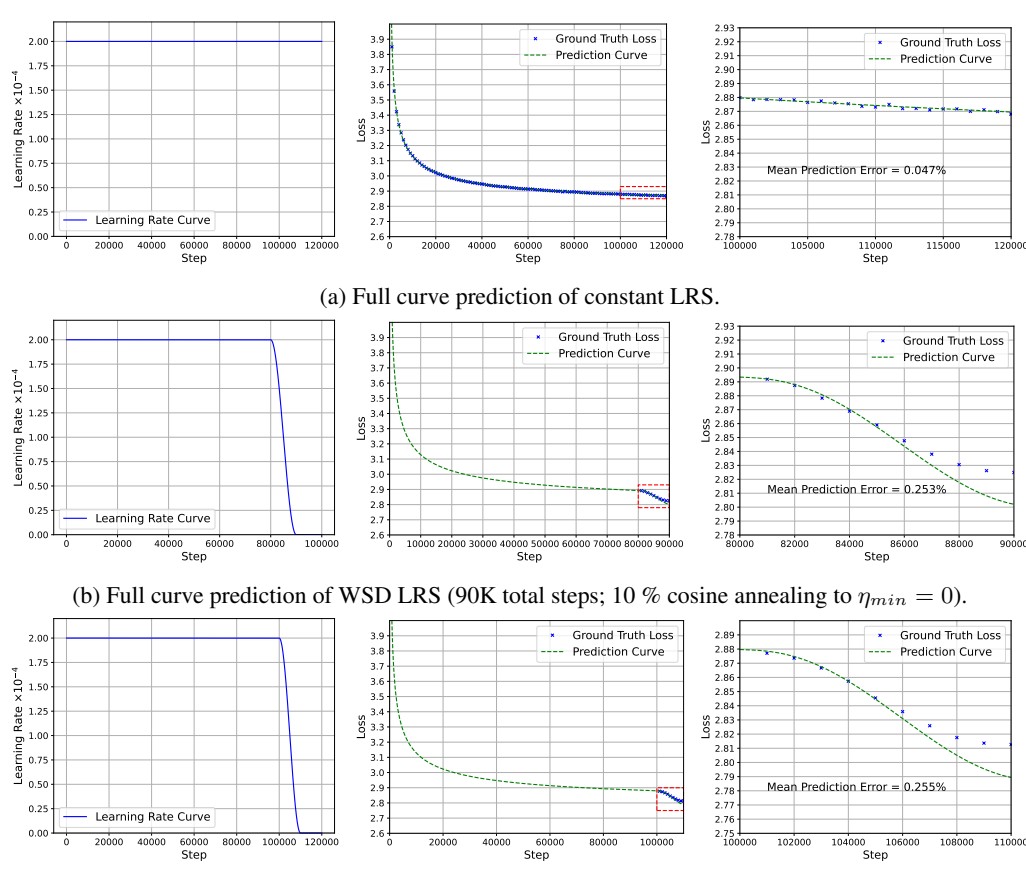

(a) Full curve prediction of constant LRS.

(b) Full curve prediction of WSD LRS (90K total steps; 10 % cosine annealing to $\eta_{min} = 0$).

(c) Full curve prediction of WSD LRS (110K total steps; 10 % cosine annealing to $\eta_{min} = 0$).

Figure 13: Full loss curve **prediction** (120K steps) by the universal loss curve equation across various LRS, fitted in Fig. 12. The left, the medium, and the right figures in each row are learning rate curve, zoomed-out loss prediction, and zoomed-in loss prediction, respectively. The red rectangle means the zoomed-in zone. The figures omit the warmup in the first 100 steps. Please note that these are predictive results, which means that none of the points in this figure (except constant LRS) are involved in the fitting process. The mean prediction errors across various LRS are low to $\sim 0.2\%$. Refer to setting B in Table 3 for experimental setups.

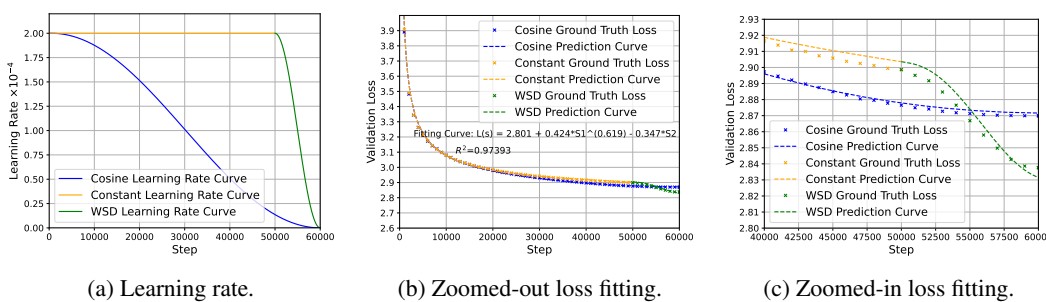

(a) Learning rate.   (b) Zoomed-out loss fitting.   (c) Zoomed-in loss fitting.

Figure 14: Full loss curve **fitting** on MoE model. After fitting, we get a **universal** loss equation $L = 2.801 + 0.424 \cdot S_1^{-0.619} - 0.347 \cdot S_2$. Refer to setting D in Table 3 for experimental setups.

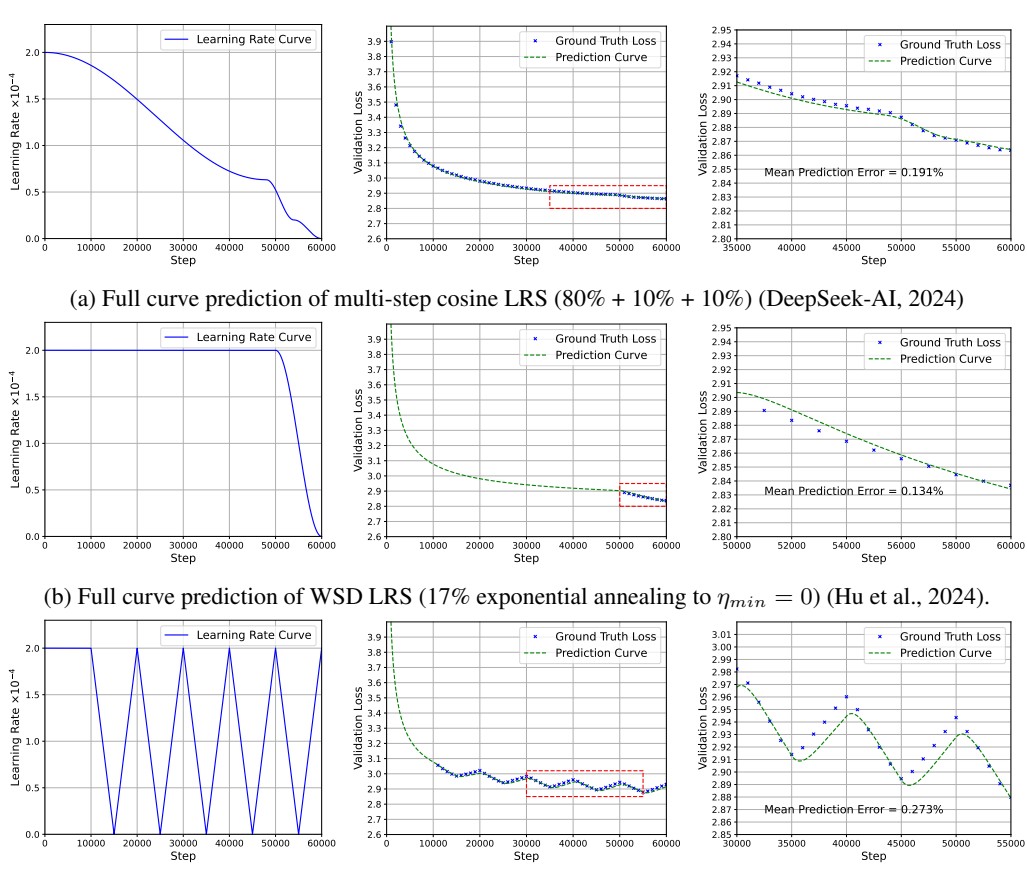

(a) Full curve prediction of multi-step cosine LRS (80% + 10% + 10%) (DeepSeek-AI, 2024)

(b) Full curve prediction of WSD LRS (17% exponential annealing to $\eta_{min} = 0$) (Hu et al., 2024).

(c) Full curve prediction of Cyclic LRS (Smith, 2017).

Figure 15: Full loss curve **prediction** on MoE model by the universal loss curve equation across various unseen LRS fitted in Fig. 14. The left, the medium, and the right figures in each row are learning rate curve, zoomed-out loss prediction, zoomed-in loss prediction, respectively. The red rectangle means the zoomed-in zone. The LR curve figures omit 500 warmup steps. Note that these are all predictive results, and none of the points in the figures are involved in the fitting process. The mean prediction errors across various LRS are low to $\sim 0.2\%$. Refer to setting D in Table 3 for experimental setups.

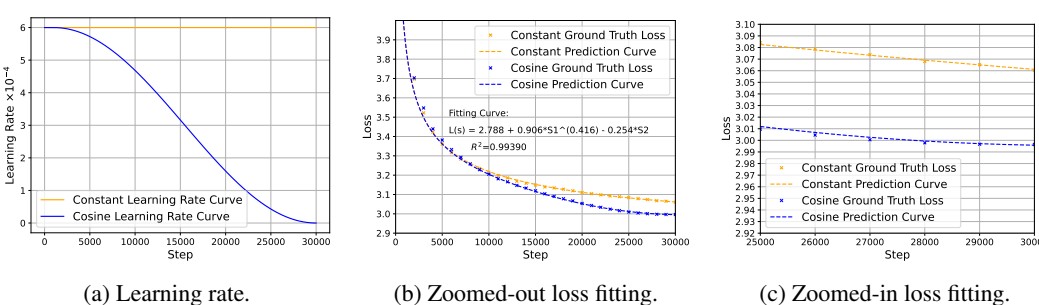

(a) Learning rate.     (b) Zoomed-out loss fitting.     (c) Zoomed-in loss fitting.

Figure 16: Full loss curve **fitting** on 30K Steps. After fitting, we get a **universal** loss equation $L = 2.788 + 0.906 \cdot S_1^{-0.416} - 0.254 \cdot S_2$. Refer to setting E in Table 3 for experimental setups.

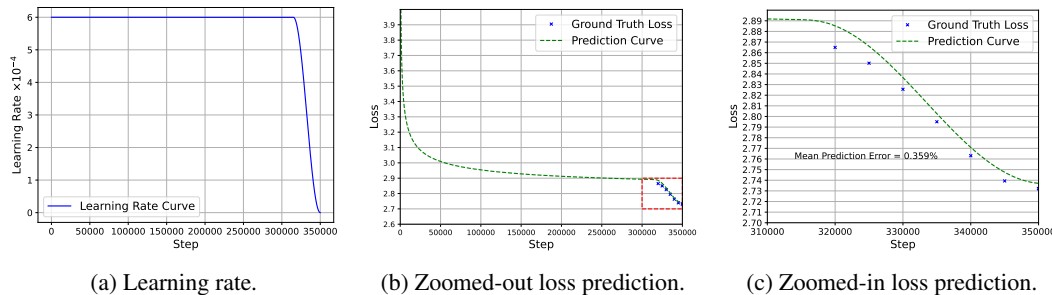

(a) Learning rate.     (b) Zoomed-out loss prediction.     (c) Zoomed-in loss prediction.

Figure 17: Full loss curve **prediction** (350K steps) by the universal loss curve equation under WSD LRS (10% cosine annealing ratio to $\eta_{min} = 0$). We adopt our equation and accurately predict the loss curve in the annealing stage after the 10x longer steps. This is meaningful to the development for large-scale LLM pre-training. Refer to setting E in Table 3 for experimental setups.

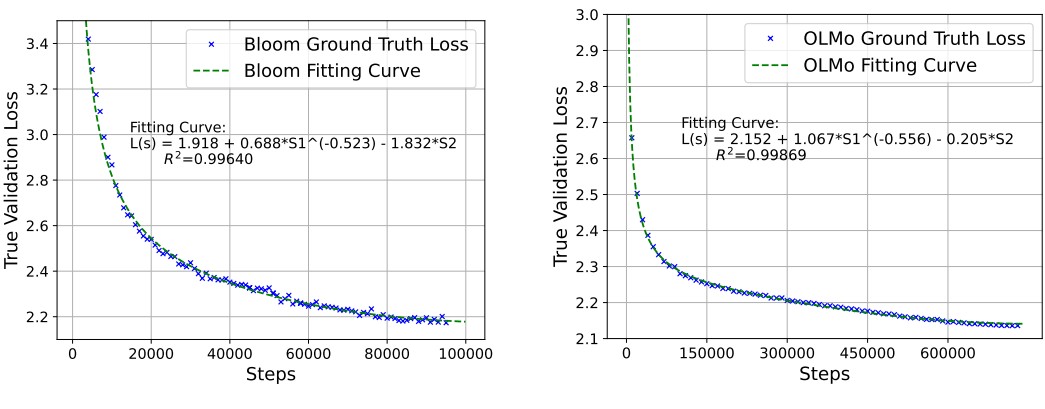

(a) Full loss curve fitting on BLOOM-176B.     (b) Full loss curve fitting on OLMo-1B (2T tokens).

Figure 18: Open-sourced full loss curve fitting using our proposed equation, which shows that our equation has strong scalability on model size and token number. We extract the curve of BLOOM from `https://huggingface.co/bigscience/bloom/tensorboard`, and we choose the column `lm-loss-validation/valid/lm loss validation` as validation loss. We extract the curve of OLMo from `https://wandb.ai/ai2-llm/OLMo-1B?nw=nwuserdirkgr`, and we choose the column `eval/pile/CrossEntropyLoss` as validation loss. Both models adopt cosine LRS.

Table 4: Ablation studies on $S_1$ and $S_2$ in our scaling law formualtion. The prediction errors on different LRS for each form are reported.

| Scaling Law Forms | Cosine | Multi-step Cosine | WSD | Cyclic |
|---|---|---|---|---|
| $L(s) = L_0 + A \cdot S_1^{-\alpha} - C \cdot S_2$ | **0.159%** | **0.176%** | **0.235%** | **0.322%** |
| w/o $S_1$: $L(s) = L_0 + A \cdot s^{-\alpha} - C \cdot S_2$ | 0.465% | 0.458% | 1.162% | 1.139% |
| w/o $S_2$: $L(s) = L_0 + A \cdot S_1^{-\alpha}$ | 1.261% | 1.265% | 1.519% | 1.203% |

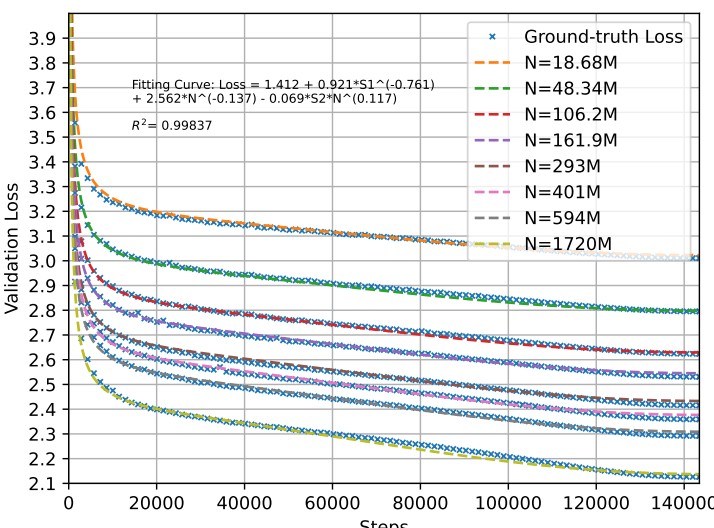

Figure 19: Curve fitting on cosine LRS (143K steps to $\eta_{min} = 0$) of many model sizes using our scaling law extended to model size $N$. Refer to setting C in Table 3 for experimental setups.

### D.6 Our $N$-extended Scaling Law on Another Experiments Setups

Fig. 7b show that our $N$-extended equation can work very well on our main experimental setup. Similarly, for proving that our $N$-extended scaling law can apply to different (but given) experimental settings, we change the setting from $A$ to $C$ (refer to Table 3) and observe whether our equation can still work or not. The fitting results are shown in Fig. 19. The results suggest that our $N$-extended scaling law with LR annealing can still work well across different experimental setups.

## E Comparison with Chinchilla Scaling Law

### E.1 Reduction to Chinchilla Scaling Law

We have proved that our scaling law can be reduced to chinchilla scaling law for constant LRS in Sec. 5. For other learning rate schedulers, we adopt a method based on statistics to show that our scaling law function can be reduced to the chinchilla scaling law. Specifically, we check whether chinchilla scaling law fits well the endpoints of loss curves predicted by our scaling law. The parameter tuple of our equation is $(L_0, A, C, \alpha)$. We then randomly sample 1000 sets of parameter tuples in some uniform distributions: $L_0 \sim U(1, 3)$, $A \sim U(0.3, 0.5)$, $C \sim U(0.2, 0.6)$, $\alpha \sim U(-0.6, -0.4)$. Each parameter tuple could be seen as the fitting result of a distinct set of experimental setups [3] (e.g. dataset, batch size, model size, etc.). For each generated parameter tuple, we apply our equation to predict the final loss of different total training steps on two LRS including cosine and WSD (10% annealing ratio). range from 5K steps to 60K steps. We conduct the prediction on two LRS including cosine and WSD (10% annealing ratio). The predicted final loss points

---

[3]It's worth noting that some of these sampled parameter tuples might not be reasonable or likely to happen in real-world scenarios, but we choose to keep them nonetheless.

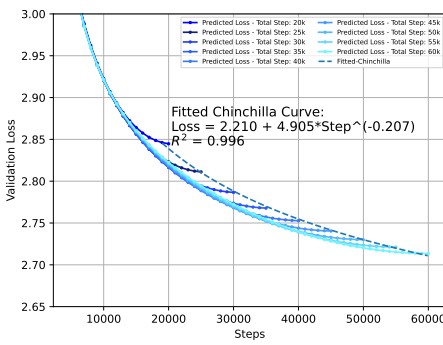 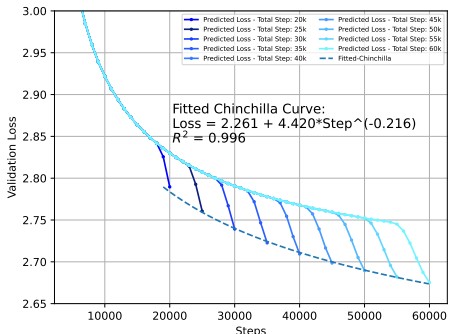

(a) The predicted loss of different total steps with cosine LRS and the fitted chinchilla curve.

(b) The predicted loss of different total steps with WSD LRS and the fitted chinchilla curve.

Figure 20: Chinchilla scaling law fits well the validation loss endpoints predicted by our formulation, taking cosine LRS (on the left) and WSD LRS (on the right) as examples.

are used to fit the chinchilla equation through minimizing the Huber loss. The fitting examples are shown in Fig. 20.

### E.2 SCALING LAW COMPUTATIONAL COST COMPARISON

We suppose a scenario where it requires 100 fitting points to get the parameters of scaling laws. We assume the distance between each point as $K$ steps. We compute the required training steps using different approaches as follows:

- Adopting Chinchilla scaling law, typical cosine LRS requires total steps of at least $1K + 2K + 3K + \cdots + 100K = 5050K$;

- Adopting Chinchilla scaling law, WSD LRS (notating annealing ratio as $r$) requires total steps of at least $(1K + 2K + 3K + \cdots + 100K)r + 100K(1 - r) = (100 + 4950r)K$.

- Adopting our scaling law, all we need is only one training curve with moderate total steps (and the number of fitting points is far more than 100), such as one curve with $50K$ steps [4]

## F  WSD SCHEDULER AND ANNEALING FUNCTIONS

Hu et al. (2024) proposes a warmup-stable-decay (WSD) LRS including three learning rate stages, which could help get a lower validation loss compared to the typical cosine LRS. The format is like

$$WSD(s) = \begin{cases} \frac{s}{T_{\text{warmup}}}\eta_{max}, & s \leq T_{\text{warmup}} \\ \eta_{max}, & T_{\text{warmup}} < s \leq T_{\text{stable}} \\ \eta_{min} + f(s) \cdot (\eta_{max} - \eta_{min}), & T_{\text{stable}} < s \leq T_{\text{total}} \end{cases} \quad (7)$$

Where $0 \leq f(s) \leq 1$ is typically a decreasing function about step $s$, and $\eta_{max}$ is the maximal learning rate. Hägele et al. (2024) consolidates the effectiveness of WSD scheduler by many empirical experiments. Moreover, Hägele et al. (2024) also finds that using 1-sqrt annealing and a moderate annealing ratio (e.g. 20%) can further decrease the final loss. The 1-sqrt annealing is defined as:

$$f(s) = 1 - \sqrt{\frac{s - T_{\text{stable}}}{T_{\text{total}} - T_{\text{stable}}}} \quad (8)$$

---

[4]The empirical rule that more fitting points always achieve better fitting results always holds true. Our equation can also use more points and LRS for fitting, such as $30K$ constant + $70K$ cosine. Nevertheless, we can collect far more fitting points than the typical scaling law with significantly fewer training steps.

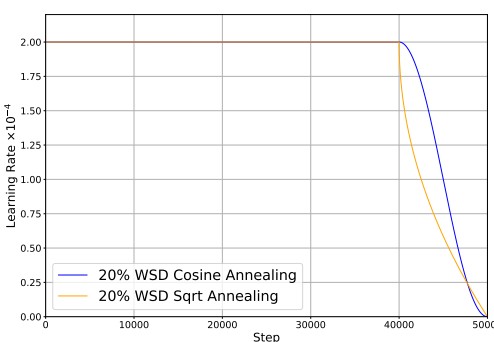 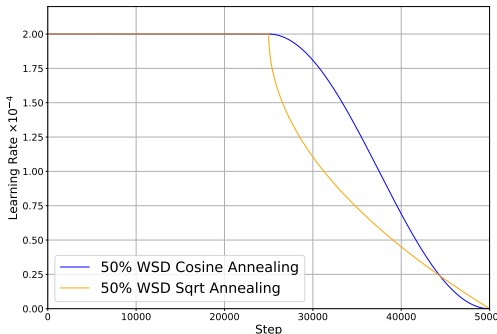

(a) LR curve of WSD (20% 1-sqrt/cosine annealing).     (b) LR curve of WSD (50% 1-sqrt/cosine annealing).

Figure 21: The learning rate curves of 20% (left) and 50% (right) annealing ratio in WSD LRS, with cosine and 1-sqrt annealing method.

Also, Hägele et al. (2024) mentions 1-square annealing method as a baseline, which is defined as:

$$f(s) = 1 - \left( \frac{s - T_{\text{stable}}}{T_{\text{total}} - T_{\text{stable}}} \right)^2 \tag{9}$$

We draw the learning rate curve of WSD (20% and 50% 1-sqrt annealing) in Fig. 21, compared with cosine annealing. Other than 1-sqrt annealing,

# G  TAKEAWAYS: EXPERIMENTAL FINDINGS VERIFICATION AND EXPLANATION

## G.1  IT VERIFIES AND EXPLAINS WHY LOSS DROPS MORE SHARPLY WHEN LR ANNEALS.

We adopt our equation to help researchers understand why loss drops more sharply when LR anneals, which has been widely observed in many previous studies. We substitute the fitted parameters (see Fig. 2) to our equation as an instance. We draw how the $S_1$-item ($A \cdot S_1^{-\alpha}$) and the negative $S_2$-item ($-C \cdot S_2$) impacts the loss along with a WSD scheduler. Fig. 22 suggests that starting from annealing stage, negative $S_2$-item has a much more significant impact on the overall loss than $S_1$-item, which makes loss drop more sharply compared with the stable LR stage. In conclusion, LR annealing brings out quick increase of the annealing area, resulting in a drastic decrease in validation loss.

## G.2  IT VERIFIES AND EXPLAINS THE PHENOMENON, WHERE CONSTANT LRS GETS A LOWER LOSS THAN COSINE LRS IF SETTING SMALL TOTAL STEPS, AND VICE VERSA.

In the experiments, we find that if we set small total steps, the final loss of constant LRS could be even lower than cosine LRS, and vice versa. Refer to the ground-truth loss in Fig. 2 (20K steps). To validate this phenomenon, we use our equation to draw the prediction loss curve of 10K total steps and 100K total steps in Fig. 23. It shows that our proposed equation can verify well that the better LRS changes over the total steps. Moreover, Fig. 23c shows the predicted final loss of different total steps using constant and cosine LRS. It further convincingly suggests that constant LRS indeed gets a lower loss if setting small total steps, but the scaling slope is smaller than cosine LRS's, resulting in higher loss in more steps.

From a more essential and comprehensive perspective, $|\frac{\partial L}{\partial S_1}|$ is a power-law decreasing function while $|\frac{\partial L}{\partial S_2}|$ is stable over training steps. In the early stages, $|\frac{\partial L}{\partial S_1}|$ is large when $S_1$ is small, thus increasing $S_1$ by maintaining large LR (e.g. constant LRS) in the early stages can greatly help reduce the loss. That is, $S_1$ plays a dominant role over $S_2$. In the later stages, $|\frac{\partial L}{\partial S_1}|$ is much smaller when $S_1$ becomes large, thus increasing $S_1$ in the later stages does not significantly help reduce the loss. That is, $S_2$ plays a dominant role over $S_1$. At this stage, It is time to start LR annealing to increase $S_2$. Interestingly, this perspective aligns directly with the idea of WSD LRS (Hu et al., 2024): In the

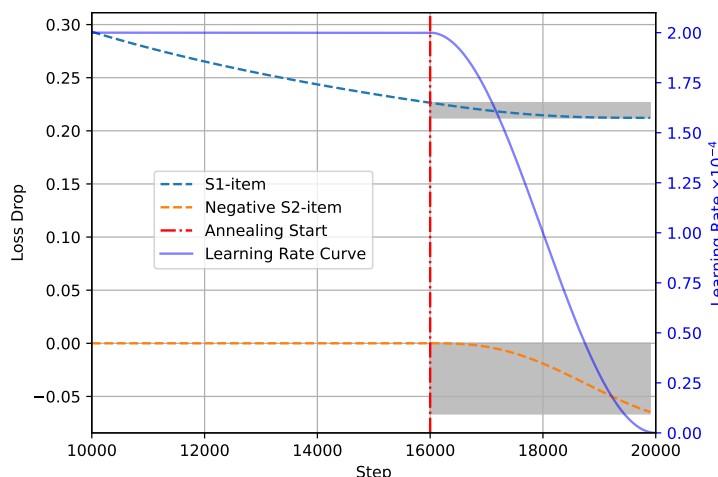

Figure 22: How S1-item and negative $S_2$-item changes in a WSD scheduler. Gray area means the amount of loss drop brought by $S_1$ and $S_2$ in annealing stage.

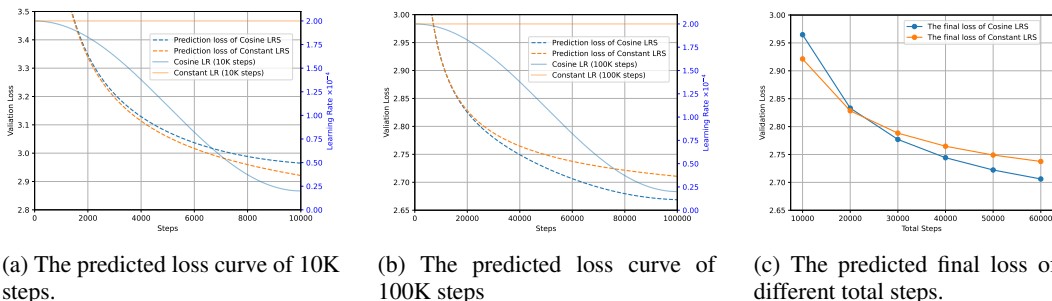

(a) The predicted loss curve of 10K steps.

(b) The predicted loss curve of 100K steps

(c) The predicted final loss of different total steps.

Figure 23: Comparison of constant and cosine LRS in different steps.

early stages, the neural network is exploring globally and it is a suitable time to use a larger LR; In the later stages, the neural network is exploring locally and it is a suitable time to use a smaller LR.

### G.3 IT VERIFIES AND EXPLAINS THAT THE OPTIMAL ANNEALING FUNCTION IN WSD LRS DEPENDS ON THE ANNEALING RATIO.

In the context of the WSD LRS, the selection of the annealing method in the annealing stage is also pivotal to optimize the training process. Hägele et al. (2024) conclude that the 1-sqrt annealing (refer to Appendix F for 1-sqrt function and curve) yields a lower final loss compared to the other annealing methods (e.g. cosine). They claim that the conclusion holds true across different annealing ratios.

However, as we predict using our equation (Fig. 24a), the 1-sqrt annealing approach does get a lower loss than the cosine annealing approach when using small annealing ratios (e.g. 10%), but it performs much worse than the cosine annealing approach when using 50% annealing ratio.

To verify whether the predictions from our equation are accurate, we conduct experiments by training models using different annealing methods and ratios within a fixed 50K total steps. As illustrated in Fig. 24b, at a 10% annealing ratio, the 1-sqrt method outperforms the cosine method, whereas at a 50% annealing ratio, the latter method exhibits a lower final loss. The true experimental results align quite well with our prediction, which also overturns some of the conclusions made by previous works. We conclude that the optimal annealing function in WSD LRS depends on the annealing ratio.

Our scaling law function provides an explanatory framework for these observations. We draw the LR curves of 1-sqrt and cosine annealing in Appendix F. At 10% annealing ratio, although the forward

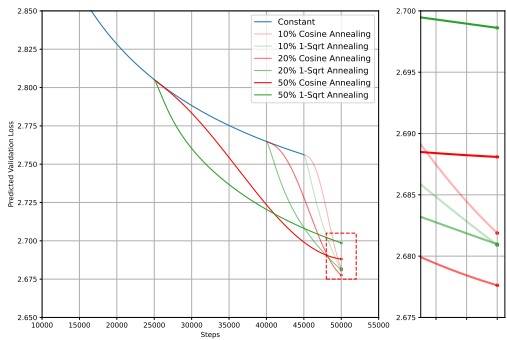 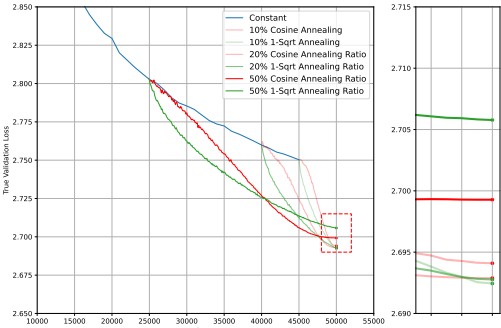

(a) The **predicted** loss curve of cosine and 1-sqrt annealing method of different annealing ratio.

(b) The **true** loss curve of different annealing ratios with cosine and 1-sqrt annealing methods.

Figure 24: The predicted (left) and true loss (right) of cosine and 1-sqrt annealing method at different annealing ratios. Experimental results (right), aligned with our prediction (left), refute the previous finding "the order and results of different annealing hold across settings" (Hägele et al., 2024).

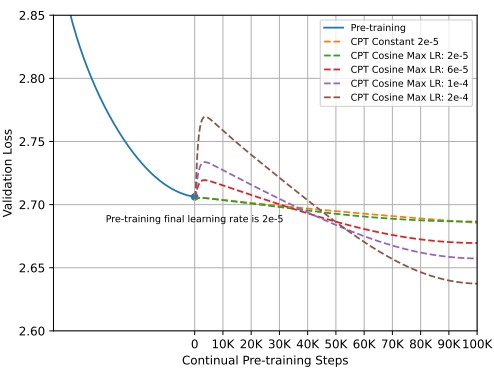 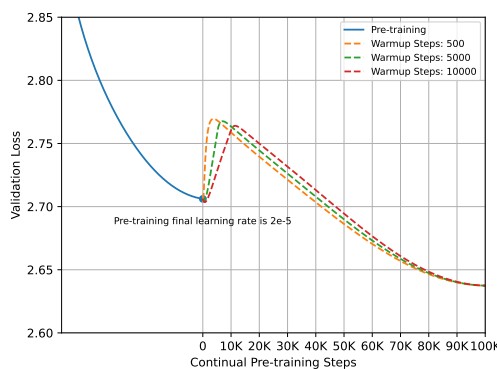

(a) The predicted validation loss with different re-warmup max LR in the continual pre-training process. All the re-warmup steps are 500 steps.

(b) The predicted validation loss with different re-warmup max learning rate in the continual pre-training process.

Figure 25: The predicted validation loss with different re-warmup max learning rate and re-warmup steps in the continual pre-training process. The LRS of continual pre-training is cosine ($T$=100K) and the min learning rate is 0.

area $S_1$ of the cosine method is slightly larger than that of the 1-sqrt method, the larger annealing area $S_2$ of the 1-sqrt method plays a more critical role in reducing the overall final loss. However, as the annealing ratio increases, the difference of $S_1$ between two LRS gradually becomes larger and larger, till breaking the delicate balance between $S_1$ and $S_2$ at 50% annealing ratio, resulting in a lower final loss for the cosine method. This relationship underscores the importance of carefully selecting the annealing strategy to optimize model training outcomes within the WSD scheduler. Still, our equation can help predict a better annealing method without experiments, which saves a lot of resources.

### G.4 IT VERIFIES AND EXPLAINS THAT IN CONTINUAL PRE-TRAINING, THE HIGHER MAX LEARNING RATE TO RE-WARMUP, THE HIGHER THE INITIAL PEAK LOSS WILL BE, AND THEN THE MORE SHARPLY IT WILL DECREASE.

In continual pre-training (CPT), the learning rate scheduler is usually set as re-warmup to a new max LR at the beginning. By many experiments, Gupta et al. (2023) concludes that the higher max learning rate to re-warmup, the higher the initial peak loss will be, and then the more sharply it will decrease.

According to our scaling law function [5], in the re-warmup process, the annealing area $S_2$ will reduce to a negative value ($S_2 < 0$) and thus the validation loss increases. The higher max LR in re-warmup, the annealing area $S_2$ becomes more negative and thus there would be a higher peak loss. But still, higher max LR could make the forward area $S_1$ grow faster and the loss decreases more sharply after re-warmup. We use the fitted equation to predict the continual pre-training process with different max LR as shown in Fig. 25a. The predicted loss curves reproduce a quite similar phenomenon with previous works (Gupta et al., 2023).

There is a more profound strategy using our equation in CPT. As shown in Fig. 25a, after ensuring total steps during CPT, we can apply our equation to predict a better max LR and scheduler to get the lowest final loss without experiments, which saves a lot of resources.

### G.5 IT VERIFIES AND EXPLAINS THAT IN CONTINUAL PRE-TRAINING, THE STEPS OF RE-WARMUP HAVE LITTLE IMPACT ON THE FINAL LOSS.

Meanwhile, how many steps to re-warmup is another important issue in the continual pre-training. Gupta et al. (2023) find that the longer re-warmup steps could smooth the transition of loss curve but the number of re-warmup steps does not significantly influence the final validation loss. We use the fitted equation to predicted the continual pre-training dynamics with different re-warmup steps. The results, shown in Fig. 25b, present a good alignment with previous works (Gupta et al., 2023).

Based on our theory, given the fixed max LR, when the re-warmup steps are longer, the annealing area decreases more slowly and the loss curve rises more smoothly, but both final $S_1$ and $S_2$ are quite stable across different re-warmup steps. First, the annealing area $S_2$ of different re-warmup steps are very close due to the same max LR and the same min LR. Besides, though different re-warmup steps bring in temporary distinct losses, re-warmup only cover a small percentage compared with all training steps. Thus, the forward area $S_1$ is also close across different re-warmup steps, resulting in the close overall loss across different steps of re-warmup.

## H DISCUSSION

### H.1 THE IMPACT OF DECAY FACTOR $\lambda$

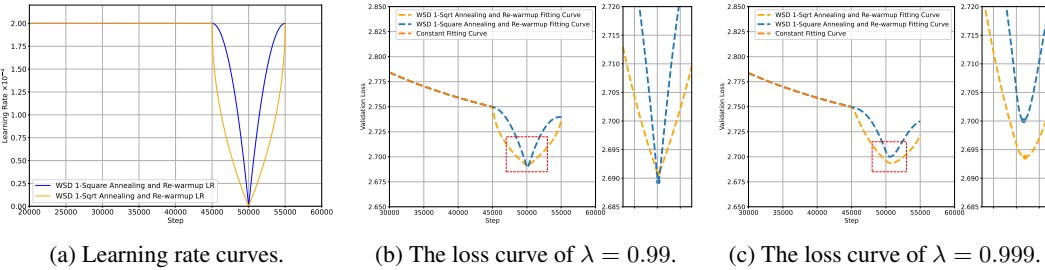

(a) Learning rate curves.        (b) The loss curve of $\lambda = 0.99$.        (c) The loss curve of $\lambda = 0.999$.

Figure 26: The comparison of fitting effect of different decay factor $\lambda$.

The decay factor $\lambda$ in our equation plays a crucial role to indicate the information retaining degree in LR annealing. We set $\lambda$ as 0.999 in our all experiments. We explore the difference from another decay factor $\lambda = 0.99$. We fit and get different equations for different $\lambda$. We compare them (1) on the predicted loss curves for 1-square and 1-sqrt annealing methods, and (2) on the predicted loss curves in different annealing ratios of WSD LRS (cosine annealing).

The results, illustrated in Fig. 26 and 27, reveal several key insights into the impact of decay factor:

**Delay Steps.** A larger decay factor results in longer delay steps. Comparing Fig. 26b and Fig. 26c, $\lambda = 0.999$ introduces a more obvious delay phenomenon, which is consistent across both the 1-

---

[5]Strictly speaking, continual pre-training process often include LR re-warmup as well as data distribution shift. Here we primarily research on the condition where there is no distribution shift between two training stages. The conclusions transfer across most cases because the loss change brought by LR re-warmup is significantly larger than the loss change brought by data distribution shift (Gupta et al., 2023; Ibrahim et al., 2024).

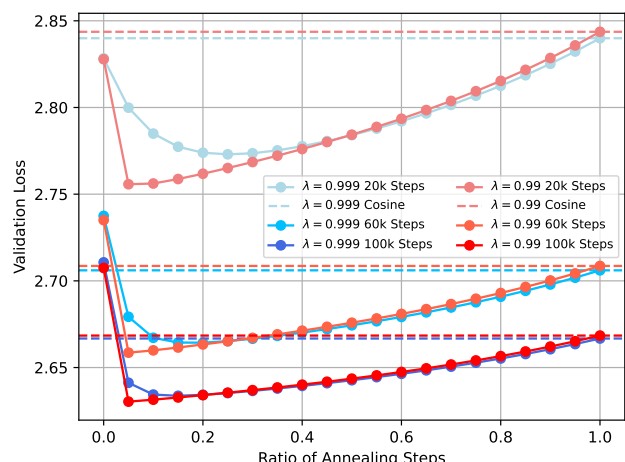

Figure 27: The predicted loss in different annealing ratios of WSD LRS for $\lambda = 0.99$ and $\lambda = 0.999$.

square and 1-sqrt annealing methods. The root reason is simple: larger $\lambda$ can retain more LR historical momentum, causing longer delay steps after LR finish annealing.

**Optimal Annealing Ratio.** a larger decay factor tends to favor a higher annealing ratio. As shown in Fig. 27, The optimal annealing ratio of $\lambda = 0.999$ is larger than that of $\lambda = 0.99$. Meanwhile, due to the similar reason, $\lambda = 0.999$ favors 1-sqrt annealing method while $\lambda = 0.99$ favors 1-square annealing method, as shown in Fig. 26.

**Balance Point between $S_1$ and $S_2$.** More essentially, the selection of $\lambda$ decides the balance point of $S_1$ and $S_2$. For example, $\lambda = 0.999$ means that, LR annealing only retain the information of previous approximately $\frac{1}{1-\lambda} = 1000$ steps, which can be seen as the window size of LR annealing momentum. The window size could be very close to the optimal annealing steps. After reaching window size, $S_2$ increases very slowly, with the cost of large decrease of $S_1$.

The analyses above highlights the importance of selecting a decay factor that aligns closely with empirical data to ensure the accuracy of predictions. We recommend that the future developers try different $\lambda$ for their own setups [6].

### H.2 POSSIBLE ROOT REASONS OF DELAY PHENOMENON IN LEARNING RATE ANNEALING

In Sec. 3, we discover the delay phenomenon, which proves that LR annealing has momentum. We discuss possible root reasons of the phenomenon in this section.

**Adam Optimizer? No.** We notice that Adam optimizer (Kingma & Ba, 2015) also has the first-order momentum decay factor $\beta_1$ and the second-order momentum decay factor $\beta_2$, which presents the possible connection to the the delay phenomenon.

We keep $\beta_1 = 0.9$, and conduct delay experiments on different $\beta_2 \in \{0.95, 0.99, 0.999\}$ (default: 0.95) of AdamW optimizer (Loshchilov & Hutter, 2017) to observe whether larger $\beta_2$ causes a more longer delay steps. The learning rate and ground-true loss curve are shown in Fig. 28a. It suggests that the ground-truth loss curves of different $\beta_2$ almost coincide with each other, and their delay steps are also the same. Therefore, we believe that Adam optimizer has little to do with the delay phenomenon, despite its momentum form seeming very related to our experiments. Speaking of which, we even once tried to mimic the form of Adam Optimizer to describe LR annealing momentum, attempting to discover a connection between them, but the fitting results were a mess.

**Forward Area $S_1$? Not Really.** No matter how LR changes, $S_1$ is always increasing over steps, resulting in consistently reducing the validation loss brought from $S_1$. Therefore, the forward area,

---

[6] Actually, $\lambda$ can be fitted as a parameter, instead of a hyper-parameter requiring manual tuning. We regard $\lambda$ as a hyper-parameter because $\lambda = 0.999$ performs well in our all experiments. Besides, fitting with $\lambda$ could bring in additional time complexity due to the recomputation of $S_2$.

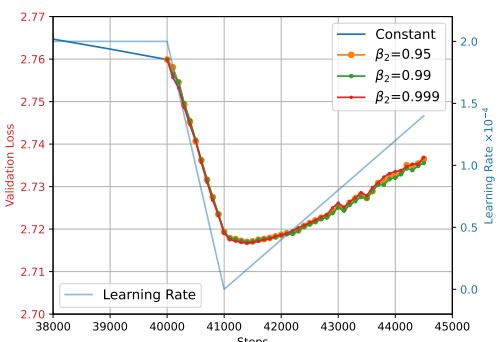 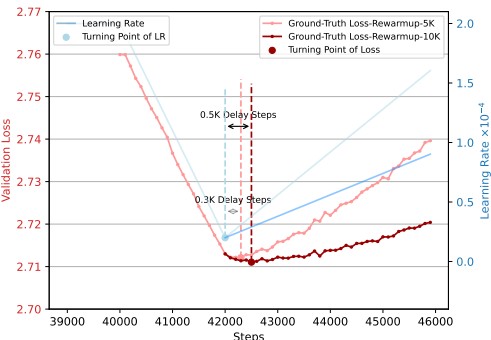

(a) The comparison of true loss curve with setting different $\beta_2$ of Adam optimizer.

(b) The comparison of delay steps of different re-warmup steps (and thus different $S_1$).

Figure 28: The possible root reason analysis (left: Adam optimizer, right: $S_1$) of delay phenomenon.

$S_1$ would lengthen delay steps in LR annealing then re-warmup, but would shorten delay steps in LR re-warmup then annealing. The delay phenomenon is indeed related to $S_1$.

But still, $S_1$ is not all the reasons of delay phenomenon. We prove this by Fig. 5b, which suggests that even though in LR re-warmup then annealing, the delay phenomenon, while not that pronounced, still exists. Moreover, we conduct delay experiments by adjusting the slope of LR after tuning point of LR. As shown in Fig. 28b, We find that more smooth slope of LR re-warmup, with smaller $S_1$, but still causes longer delay steps. Therefore, we conclude that $S_1$ indeed influences the specific delay length, but is not the root reason.

**Other Possible Reasons?** The delay phenomenon could be intuitive in some cases. For example, suppose that learning rate decreases directly from 2e-4 to 2e-5 in one step, and maintains 2e-5. In this case, although the loss would decrease to a lower value but the parameter changes in one step is too small in neural networks. Given a sudden low LR, neural networks still require some steps to gradually optimize to a local minimum, incurring delay phenomenon. But still, analysis above still ends with a rough description, and we have not figured out the root reasons and look forward to completing this part in future work.

## H.3  OTHER POSSIBLE SCALING LAW FORMATS WITH LR ANNEALING

**Adding a LR-weighted Coefficient to $S_2$?** Imagine that when LR anneals to nearly $0$, the neural network's parameters almost do not change and the validation loss should not change, either. However, as defined in our equation, Eq. 1, $S_2$ still has historical momentum even if LR is nearly $0$, making the loss continue to decrease and misalign with observed training dynamics.

To cover this corner case, we try a revision to our equation and add a LR-weighted coefficient to $S_2$. Specifically, we adjust $S_2$ to more approach $0$ when $\eta$ is close to $0$, counteracting the original formulation's tendency to overestimate loss reduction when $\eta \approx 0$.

The revised equation for the annealing area $S_2$ in our scaling law function is as follows:

$$
\begin{aligned}
m_i &= \lambda \cdot m_{i-1} + (\eta_{k-1} - \eta_k) \\
&= \sum_{k=1}^{i} (\eta_{k-1} - \eta_k) \cdot \lambda^{i-k} \\
S_2 &= \sum_{i=1}^{s} m_i \cdot \eta_i^{\epsilon}
\end{aligned}
\tag{10}
$$

Where the red part is the added LR-weighted coefficient and $\epsilon$ is a undetermined positive constant. $\epsilon$ could be very small in practice.

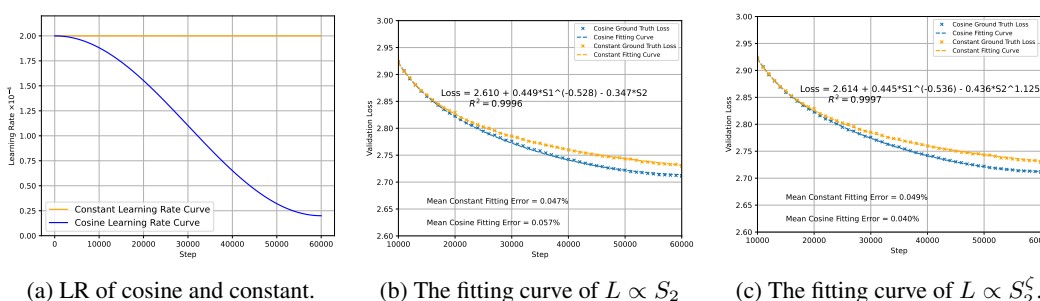

(a) LR of cosine and constant.  (b) The fitting curve of $L \propto S_2$  (c) The fitting curve of $L \propto S_2^\zeta$.

Figure 29: The comparison of fitting effect between $L \propto S_2^\zeta$ with $L \propto S_2$.

We have tried the revised function to fit data. We find that the fitting results are quite similar and $\epsilon$ is very close to 0, showing little use in practical effect. Hence, we adopt the original format in our experiments [7].

$L \propto S_2^\zeta$ **rather than** $L \propto S_2$**?** Actually, all we know is that $L$ and $S_2$ have a positive correlation. Thus $L \propto S_2^\zeta$ rather than $L \propto S_2$ might be a more reasonable assumption. That is, our equation would be changed to $L(s) = L_0 + A \cdot S_1^{-\alpha} - C \cdot S_2^\zeta$. Theoretically, the introduction of $\zeta$ as an additional fitting parameter is expected to provide a more nuanced control over how changes in the learning rate annealing affect validation loss, potentially leading to improve the accuracy of our equation.

However, the empirical results, as depicted in Fig. 29, demonstrate that the fitting improvement with the inclusion of $\zeta$ is quite marginal when compared to the version without this parameter. This slight enhancement does not justify the additional complexity introduced by managing negative values of $S_2$. Furthermore, the empirical observation that $\zeta$ tends to converge close to 1 (e.g. 1.125 in Fig. 29c) reinforces the notion that the original formulation of the function, without the power term $\zeta$, is adequately robust. This finding suggests that the direct influence of the learning rate annealing area, as initially modeled, sufficiently captures the essential dynamics without the need for this additional complexity. Another additional complexity lies in that $S_2^\zeta$ becomes incalculable when $S_2 < 0$ in LR re-warmup.

Studies of scaling laws are mostly empirically driven. Over-parameterizing the scaling law equation essentially leads to more accurate fitting results. However, it will also complicate the final format and hinders us to focus on major factors for the training dynamics. We choose our main format not due to absolute prediction accuracy but to pursue the simplification (i.e., fewest extra parameters) to model the essential training dynamics of LLMs. Notably, in our main format in Eq. 1, we only introduce one extra parameter compared to Chinchilla scaling law, i.e., the coefficient $C$ of the $S_2$ term. As suggested in Sec. 3, our main scaling law format still has a strong and robust capacity across many practical scenarios. We believe and expect that there should be more powerful specific format (maybe with more parameters) after this work.

### H.4 Optimizing Learning Rate Schedule

A natural next step of this work would be optimizing LR schedule based on our proposed scaling law. From a practical engineering aspect, it is feasible and efficient to select better LRS from many candidates based on the prediction of the scaling law. Specifically, WSD (rather than cosine) LRS should be used to confirm the larger values for both $S_1$ and $S_2$, as stated in Sec. 4.2 and Appendix G.2; WSD LRS annealing ratio can be determined by the method stated in Sec. 4.3; Annealing function can be selected by the method stated in Appendix G.3. It is quite easy to get the (nearly) optimal LR schedule based on the composition of the methods above.

From another perspective, one might adopt our scaling law to directly solve optimal LR schedule mathematically. It could be found that our Eq. 1 leads to a collapsed LRS: some zero learning rates at last. This problem is related to the issue ($\eta \approx 0$ case) that we discussed above in Appendix H.3.

---

[7] We still recommend future developers to try this format if possible.

Mathematical optimization strongly depends on an absolute accuracy, while our scaling law in such scenario does not achieve perfectly 100% accuracy (shown in Fig. 3). In comparison, our mentioned variant forms (e.g., Eq. 10) with extra parameters should be more preferably used to solve the optimal LRS, because they cover more corner cases and have higher accuracy. We believe that in the future, there will be stronger and more parameterized specific forms, which are more suitable for directly mathematically solving the optimal LR schedule. At this stage, we believe that the approach based on the practical engineering mentioned above is sufficient to obtain a (nearly) optimal LR schedule.

