# OpenReview forum: "Scaling Law with Learning Rate Annealing"
_ICLR.cc/2025/Conference — Submitted to ICLR 2025_

### Official Review · Reviewer_5x7q · 2024-11-02

**Soundness:** 2
**Presentation:** 3
**Contribution:** 3
**Rating:** 8
**Confidence:** 4

**Summary:**

This paper gives an analytical expression for the loss a a function of the learning rate schedule (after fixing “training and validation dataset, the same model size, the same training hyper-parameters such as warmup steps, max learning rate ηmax and batch size”). Their common expression accounts for both the benefit in loss due to more training duration as well the benefit due to learning rate annealing.

**Strengths:**

Learning rate scheduling is an important problem in the training of LLMs, which had not yet been addressed in general (i.e. given a schedule predict the final loss), this paper addresses that problem. While I do not believe the results in the paper are perfect, they are an important first step in addressing this problem. As an example of their predictive power they are able to approximately predict the behavior of loss even when it is non-montonic (Fig 3d).

**Weaknesses:**

Weakness:
1. The results in the paper would have been much easier to interpret had there been a comparison to simple baselines. For example, how good is the fit which averages the loss curves of all the schedules tried?  (for a fixed eta_max, batch size etc.)
2. One of the main application the results is comparing different schedule. In Section 4.3 “Determining optimal annealing ratio of wsd scheduler” the authors claim to explain the commonly observed durations of annealing in WSD (10-20%). It would have been more convincing if the authors had validated their curves empirically. For example in the predicted curves

   a. The difference between cosine and WSD (with optimal hyperparams is around .05)

   b. The optimal anneleaning fraction decreases with more training.

    Are these two observations matched in practice? As an evidence against a., in Hägele 2024 they observe that after decaying cosine to 0 they can match the performance of WSD.

**Questions:**

Some questions are present in the previous section of the review.

“The results show an almost perfect fit, achieving a coefficient of determination (R2) greater than 0.999.” Is this correlation for a single fit (i.e. “Given the same training and validation dataset, the same model size, the same training hyper-parameters such as warmup steps, max learning rate ηmax and batch size) or across multiple model fits? Because fitting across multiple models would not be statistically sound, if that is the case the authors should instead report the average of R^2 across different models.

---

> ### Author Response · Authors · 2024-11-17
> **Response to Reviewer 5x7q**
>
> ## Q1: Comparison to simple baselines.  How good is the fit which averages the loss curves of all the schedules tried?
>
> We deeply appreciate the time and effort you have put into reviewing our paper and providing valuable feedback. As you commented, we are the first the propose a scaling law to model the full loss curve. Therefore, there is no other baselines to compare with. Still, we would like to provide some simple baselines (ablation studies) to more comprehensively reflect the performance of our law. In each form, we re-fit based on 20K cosine + constant and predict the schedules in Fig. 3.
>
> | Scaling law forms                                         | Prediction error on cosine LRS | Prediction error on multi-step cosine LRS | Prediction error on WSD LRS | Prediction error on Cyclic LRS | Mean prediction error |
> | --------------------------------------------------------- | ------------------------------ | ----------------------------------------- | --------------------------- | ------------------------------ | --------------------- |
> | Ours: $L(s) = L_0 + A\cdot S_1^{-\alpha} - C\cdot S_2$    | 0.159%                         | 0.176%                                    | 0.235%                      | 0.322%                         | 0.223%                |
> | w/o $S_1$: $L(s) = L_0 + A\cdot s^{-\alpha} - C\cdot S_2$ | 0.465%                         | 0.458%                                    | 1.162%                      | 1.139%                         | 0.806%                |
> | w/o $S_2$: $L(s) = L_0 + A\cdot S_1^{-\alpha}$            | 1.261%                         | 1.265%                                    | 1.519%                      | 1.203%                         | 1.312%                |
>
> We have also included the results in our revised paper (Table 4).
>
> ## Q2: Are the observations matched in practice in Section 4.3 ?
>
> Thanks for the valuable feedback, we indeed validate our predictions empirically.  Specifically, in Figure 24, we compare 10%, 20%, and 50% annealing ratios (in both ground-truth and prediction), The ground-truth and prediction coincide well, and we find that in our experimental setting, 50% annealing ratio is worse than 10-20% annealing ratio. Here we also present more explanations for the phenomenon you mentioned.
>
> **b. The optimal annealing fraction decreases with more training.**
>
> This is exactly the same conclusion made by Hägele' work [1]. Here we copy the statement from their paper (the caption of Fig. 6)
>
> > A long training run suggests that a small number of cooldown steps can match cosine for long training... we find that the required duration of cooldown to match the cosine loss decreases with longer training.
>
>
>
> **a. The difference between cosine and WSD (with optimal hyperparams is around .05)**
>
> We believe that the difference between cosine and WSD LRS is affected by the detailed experimental settings. For example, we present the difference between cosine and constant LRS in Fig. 23 in Appendix G.2. It is shown that if we set small total steps, the final loss of constant LRS could be even lower than cosine LRS, and vice versa.
>
> Therefore, what really matters is that our scaling law is able to predict the final loss under all kinds of circumstances. We believe that this finding is Not against the evidence of Hägele' work [1] because we have totally different experimental settings (and thus we have different $L_0, A, C, \alpha$). Assuming we fit and get the parameters for their loss curves, we believe that the fitted equation can also lead to the same conclusion with theirs. In summary, our scaling law can uncover some general patterns (e.g., the optimal annealing fraction decreases with more training.) and provide specific predictive results under some particular experimental setups (e.g., the differences among various LRS at a specific step).
>
> [1] [Scaling Laws and Compute-Optimal Training Beyond Fixed Training Durations](https://arxiv.org/pdf/2405.18392å)
>
>
>
> ## Q3: Is this correlation for a single fit or across multiple model fits?
>
> This is a **single fit**. We have labeled the $R^2$ value on all fitting plots, and each fitting plot represents a single fit, indicating that multiple loss curves are simultaneously fitted using **exactly the same set of parameters**.
>
> Finally, we thank again for your valuable feedback! If you have any other questions or concerns, we are happy to answer and continually improve our work based on your feedback.

---

> ### Author Response · Authors · 2024-11-22
> **Please let us know if you have any further concerns.**
>
> We are very grateful for the opportunity to improve our work through your valuable feedback. If you think our response has addressed your concerns and questions, we sincerely hope you might consider raising your score. Should you have any further inquiries or require additional clarification, we warmly welcome your questions and are eager to provide comprehensive responses.

---

> > ### Comment · Reviewer_5x7q · 2024-11-29
> >
> > Thank you for your response. This addresses some of my concerns.
> >
> >
> > I think while there are issues with the paper (as pointed out by other reviewers), this is the first work on an important problem. As such I have increased my score.

---

### Official Review · Reviewer_5e85 · 2024-11-03

**Soundness:** 4
**Presentation:** 3
**Contribution:** 3
**Rating:** 8
**Confidence:** 4

**Summary:**

The paper proposes a new scaling law which captures the complete loss curve of a training run, instead of only fitting the final loss as done by traditional scaling laws. It shows that just using a constant and cosine schedule curve, it can predict the loss of any other lr schedule, including re-warmup periods.

**Strengths:**

The paper proposes a new scaling law for modeling the complete training loss curve for any schedule. It shows impressive results on predicting the loss curves of various lr schedules such as WSD, re-warmup, by training only on a single constant and cosine schedule trajectory. It shows results across various architectures such as MOEs as well as various model scales.

**Weaknesses:**

Some points in section 4.4 and 6 seem irrelevant. In section 4.4, the first point, about loss decreasing suddenly at lr annealing, the given equation has an inherent term S2 modeling the same. It feels a bit redundant to state this point, and this definitely cannot be stated as 'the reason behind why loss drops suddenly'. Similarly, in section 6, the exploration of the question 'why is there a momentum in loss while lr annealing' seems insufficient.

**Questions:**

1. Is it possible for given values of lambda and other parameters, to predict the optimal learning rate schedule for a given training duration?

2. It would be really interesting to study the tradeoff in MOEs for the number of experts with different training durations - should the number of experts be increased for longer training durations?

---

> ### Author Response · Authors · 2024-11-17
> **Response to Reviewer 5e85**
>
> ## Q1: Some points in section 4.4 and 6 seem irrelevant.
>
> **In section 4.4, the first point, about loss decreasing suddenly at lr annealing**
>
> Thanks for your valuable feedback! $S_2$ is not the only factor that influence losses in LR annealing. Actually, when LR anneals, $S_1$ increases less and makes loss decrease less suddenly, while $S_2$ makes loss decrease more suddenly (see Fig. 22 for the specific trends). There is a trade-off between $S_1$ and $S_2$ and the more essential reason is that $S_2$ plays a more dominant role than $S_1$ in LR annealing. We agree that it is more empirical rather than real physics behind the phenomenon. We have revised our statements in our paper.
>
> **In section 6, the exploration of the question "why is there a momentum in loss while lr annealing" seems insufficient.**
>
> We are thankful for your thorough review and valuable comment on our work. We derive the momentum form based on the delay phenomenon in section 3.2. In section Appendix H.2, we analyzed three possible root reasons behind the delay phenomenon, including Adam optimizer, $S_1$ in our law, and a proposed simple intuition. We found that while these factors are related, they are not the root reasons. We believe this is a really important but unresolved issue that requires further investigation in the future (as we explicitly state this in the paragraph "Other Possible Reasons?" in Appendix H.2.).
>
>
>
> ## Q2: Is it possible for given values of lambda and other parameters, to predict the optimal learning rate schedule for a given training duration?
>
> **Yes!** We answer this question from two aspects: engineering and mathematics.
>
> 1. **From a practical engineering aspect**, we have indeed presented how to get the (nearly) optimal learning rate schedule in this work. Specifically, WSD LRS should be used to confirm the larger values for both $S_1$ and $S_2$, as stated in section 4.2 and Appendix G.2; WSD LRS annealing ratio can be determined by the method stated in section 4.3; Annealing function can be determined by the method stated in Appendix G.3. In practical engineering, it is quite easy to get the (nearly) optimal learning rate schedule based on the composition of the methods above.
> 2. **From a mathematical aspect**, it becomes an optimization problem, that is, searching a learning rate sequence $(\eta_1, \eta_2, \cdots, \eta_s)$, to optimize $\text{max }L(s)= L_0 + A\cdot S_1^{-\alpha} - C\cdot S_2$ . We believe that the solution can be built up by some non-linear optimization algorithms (e.g. neural networks + gradient decent). This is a quite interesting thing on which the following work can have deeper explorations. At this stage, we believe that the approach based on the practical engineering mentioned above is sufficient to obtain a (nearly) optimal learning rate schedule.
>
>
>
> ## Q3: It would be really interesting to study the tradeoff in MOEs for the number of experts with different training durations - should the number of experts be increased for longer training durations?
>
> We are glad that you are interested in the MoE part in our work. Our scaling law formulation could not directly reflect the relationship between the number of experts and training durations. However, we believe that when the number of experts changes (and controlling other experimental settings), the training loss curves will change, too, and $L_0, A, C, \alpha$ in our scaling law will change accordingly. That is to say, it is quite possible to research this interesting question by researching how  $L_0, A, C, \alpha$ changes over the number of experts. Also, it is feasible to substitute specific training durations into our scaling law to calculate the loss and compare which is better for different number of experts. We do not conduct the experiment in this work, but we believe that the our scaling law can be extended to many application scenarios (e.g. # experts v.s. training durations), and providing practical suggestions for researchers.
>
>
>
> Finally, we thank again for your valuable feedback! If you have any other questions or concerns, we are happy to answer and continually improve our work based on your feedback.

---

> > ### Comment · Reviewer_5e85 · 2024-11-28
> >
> > Thanks for the response. When I said there is insufficient exploration of the question 'why is there a momentum in loss while lr annealing', I meant something along the lines of 'the variance accumulated from previous steps takes some time to decay, when reducing the learning rate' or possibly something related to 'edge of stability'. I am unsure how to formalize it at this point.
> >
> > However, my concerns and questions have been addressed and I maintain my recommendation.

---

### Official Review · Reviewer_up2N · 2024-11-05

**Soundness:** 2
**Presentation:** 3
**Contribution:** 2
**Rating:** 5
**Confidence:** 3

**Summary:**

This paper proposes a novel scaling law for training loss of language models when annealing learning rates. The new formulation
accounts for two main effects: (1) power-law scaling over data size, and (2) the additional loss reduction during LR annealing.  Applying the scaling law with LR annealing and fitting only one or two training curves, the authors show that their scaling law can accurately
predict the loss at any given step under any learning rate scheduler, under certain restrictions. The authors provide extenisive experimental verification for their scaling law for a range of hyper-parameters and model architectures, which can even be extended to scaling effect of model sizes.

**Strengths:**

This paper is well-written and easy to understand. The topics studied in this paper,  scaling law for training loss when annealing learning rates, is very related to the topic of ICLR conference and is of great significance, both theoreitcally and empirically.  The specific scaling law (Equation (1)) seems novel to me. The authors also provide extensive experimental justification for the accuracy of the scaling law, e.g. for cosine or WSD schedules. Overall I feel like this scaling law is quite non-trivial and will shed some light on studying the optimal learning rate schedule in pretraining, at least under certain conditions.

**Weaknesses:**

It is not completely clear under what assumptions the scaling law holds. Without necessary restrictions or premises, the scaling law seems problematic for the following obvious reasons.

1. Adding more steps with zero learning rate always reduces the predicted final loss. Any reasonable scaling law should be invariant to such operations like adding steps with zero learning rate which does not change the parameters at all.
2. Multiplying the entire schedule elementwise by a positive constant larger than 1 always decreases the predicted loss. This suggests that the scaling law must break when we use a sufficiently large peak learning rate. This is a very important limitation and I could not find related discussion on this.
3. Combing 1 and 2, if I solve for optimal LR schedule, what I get is at the first step I use an infinitely large LR, then I use 0 LR for many steps. This will give me a negative loss.


Besides the above main concern, I would like to see real applications of the scaling law, e.g., how we can make training or LR tuning more efficient by using this scaling law. I also would like to see more theoretical justifications for the scaling law.

**Questions:**

See weakness

---

> ### Author Response · Authors · 2024-11-15
> **Response to Reviewer up2N [1/2]**
>
> ## Q1: Adding more steps with zero learning rate always reduces the predicted final loss.
>
> Thank your for your valuable comment!
>
> We have indeed discussed this scenario in "Section 6: Discussion" (Line 527-528) and introduced a possible variant of our equation in appendix H.3. Specifically, we can add a LR-weighted coefficient to $S_2$, where $S_2=\sum \limits_{i=1}^{s}m_i\cdot \color{red} \eta_{i}^{\epsilon}$ and $\epsilon$ is a newly added parameter. This format can well address the issue that “adding more steps with zero learning rate always reduces the predicted final loss”, because it makes the new terms of $S_2$ become zero when learning rate is $0$. We encourage other researchers to try this format (see the footnote in Appendix H.3).
>
> We choose $S_2=\sum \limits_{i=1}^{s}m_i$ as our main format for the following reasons (where we have briefly discussed in the Appendix H.3):
>
> 1. When adopting the variant format, $\epsilon$ is small after fitting in practice, which makes the whole result become very close to the original format's result (i.e. $\eta_i^{\epsilon}\approx 1$) when $\eta$ is not that extremely near zero, which shows limited use in practical effect as in most situations, the learning rate should not be (extremely near) zero.
> 2. Notably, for our main format, we have only one extra parameter, the coefficient of $S_2$ term, $C$, compared to Chinchilla scaling law. However, we get a really accurate prediction results (shown in Section 3) and show the main format is quite robust across very diverse experimental setups (e.g. training hyper-parameters, model architectures), acknowledged by reviewer `rnjo`, `up2N` and `5e85`. In general,  $S_2=\sum \limits_{i=1}^{s}m_i$ utilizes the fewest extra parameters but model the essential training dynamics in LR annealing process to the greatest extent.
> 3. Still, we do not deny the variant $S_2=\sum \limits_{i=1}^{s}m_i\cdot \eta_{i}^{\epsilon}$. When some extreme corner cases really happen (e.g., very small learning rates are really padded at last as assumed), we still recommend future researchers to try this variant format for fitting, and the results would be expected to get better.
>
>
>
>
>
>
>
> ## Q2: Multiplying the entire schedule elementwise by a positive constant larger than 1 always decreases the predicted loss.
>
> There might be a misunderstanding about our scaling law. We present the retrictions for our scaling in Line 279-282:
>
> > Given the same training and validation dataset, the same model size, the same training hyper-parameters such as warmup steps, **max learning rate $η_{max}$** and batch size, the language modeling loss at training step $s$ empirically follows the equation $L(s) = L_0 + A\cdot S_1^{-\alpha}-C\cdot S_2$, where S1 and S2 are defined in Eq. 1. $L_0$, $A$, $C$, $\alpha$ are positive constants.
>
> Our scaling law cannot predict across different max learning rates using one single set of parameter. Loss curves with different max learning rates have totally different $L_0$, $A$, $C$, $\alpha$. When multiplying the entire schedule elementwise by a positive constant larger than 1, the max learning rate also becomes larger and the original parameters, $L_0$, $A$, $C$, $\alpha$ are not suitable anymore. Instead, we should substitute specific parameters into the different equations for comparison between different $\eta_{max}$.
>
> For example, in our main experiments (Fig. 2 and Fig. 3),  we use the equation $L(s) = 2.628 + 0.429 \cdot S_1^{−0.550} - 0.411 \cdot S_2 $ for $\eta_{max}=2e-4$. We control other all conditions and only increase $\eta_{max}$ to $4e-4$ and re-train the models, then we get a different equation after fitting: $L(s) = 2.680 + 0.491\cdot S_1^{-0.639} - 0.253\cdot S_2$  ($R^2>0.999$).

---

> ### Author Response · Authors · 2024-11-15
> **Response to Reviewer up2N [2/2]**
>
> ## Q4: Real applications of the scaling law.
>
> We clarify the applications of the scaling law from two aspects:
>
> 1. **Optimizing LR schedule:** Many previous works have pointed out that LR schedule has a great impact on the final validation loss like [1], [2], [3]. We believe that Section 4, Application is well aligned with real and practical scenarios for tuning LR schedule. And reviewer `rnjo`, `5e85`, `up2N` (you) all agree that our scaling law can be applied to study the optimal learning rate schedule.  Specifically, Section 4.1 shows how to determine min lr and cycle length in cosine LRS; Section 4.2 shows how to use our scaling law to select LRS among cosine, and multi-step cosine WSD; Section 4.3 shows how to determine the annealing ratio in WSD LRS; Appendix G.3 shows how to determine the annealing function in WSD LRS ...
>
>    Notably, Among all these applications, we judge which LRS is better or worse only through our scaling law prediction, which is cost-free and really efficient.
>
> 2. **Doing exactly what Chinchilla scaling law does, but much more efficiently.** Chinchilla scaling law, $L(D) = L_0 + A\cdot D^{-\alpha}$ ($D$ denotes dataset size) has been widely used to many scenarios like predicting the final loss when consuming much more tokens. In Section 5, we compare our scaling law with Chinchilla scaling law. We show that our scaling law is a generalized form of Chinchilla scaling law and can be reduced to Chinchilla scaling law. Simply speaking, data points that follow Chinchilla scaling law, also follow our scaling law. However, Chinchilla scaling law can only adopt the final loss as fitting points while our scaling law makes use of every points including intermediate losses. Therefore, as shown in Table 2, our scaling law can do what chinchilla do, but with only <1% cost for training to collect fitting points.
>
> [1] [MiniCPM: Unveiling the Potential of Small Language Models with Scalable Training Strategies](https://arxiv.org/pdf/2404.06395)
>
> [2] [Scaling Laws and Compute-Optimal Training Beyond Fixed Training Durations](https://arxiv.org/pdf/2405.18392)
>
> [3] [DeepSeek LLM: Scaling Open-Source Language Models with Longtermism](https://arxiv.org/pdf/2401.02954)
>
>
>
> ## Q5:  More theoretical justifications.
>
> We are very glad that you are interested in the more theoretical justifications of our scaling law. We honestly admit that our scaling law is empirical, and we have claimed that our scaling law is empirical in the first line of the abstract. However, we would like to point out that even the Chinchilla and OpenAI scaling laws are, in fact, empirical, as explicitly stated in their respective papers. We fully understand your concern regarding the lack of theoretical justification, but we believe that in AI domain, theoretical proofs often follow empirical works. For example, [4] presents some reliable theoretical justifications for scaling law after three years since OpenAI scaling law paper is published. We are continuously working on the theoretical aspects and this is indeed a very important future work.
>
> Actually, surprisingly, some related works have emerged very recently [5]. Their theoretical derivations are well matched with our scaling law. If you are interested in the theoretical justifications, we would greatly appreciate it if you could kindly take the time to read it.
>
> Finally, we thank again for your valuable feedback! If you have any other questions or concerns, we are happy to answer and improve our work based on your feedback.
>
> [4] [The Quantization Model of Neural Scaling](https://arxiv.org/pdf/2303.13506)
>
> [5] [Understanding Warmup-Stable-Decay Learning Rates: A River Valley Loss Landscape Perspective](https://arxiv.org/pdf/2410.05192)

---

> ### Author Response · Authors · 2024-11-22
> **Please let us know if you have any further concerns.**
>
> We are very grateful for the opportunity to improve our work through your valuable feedback. If you think our response has addressed your concerns and questions, we sincerely hope you might consider raising your score. Should you have any further inquiries or require additional clarification, we warmly welcome your questions and are eager to provide comprehensive responses.

---

> ### Comment · Reviewer_up2N · 2024-11-23
>
> I thank the authors for their clarification on the fixed maximum learning rate. That resolves the concern raised in the original review on that point. However, I am still concerned about the undesired behavior of scaling law in case of appending zero or extremely small learning rates in the end. The discussion in the appendix doesn't really present a solution to this issue.
>
> Regarding the application of the scaling law, I am referring to predicting phenomena that we do not know or proposing novel schedules that works well. For example, I think a basic sanity check the authors should include in the paper is to exactly solve the learning rate schedule that minimizes the predicted loss, and then test the performance of the optimal schedule predicted by your scaling law, e.g., in the context of Figure 2, $L(s) = 2.628 + 0.429\cdot S_1^{−0.550} − 0.411 · S_2$. Given that $S_2$ is a linear function of the learning rates per step and $S_1$ is a function of the sum of learning rates, solving the exact optimal schedule is very easy. I will increase my score to at least 6 if the performance of the optimal schedule predicted by the authors' scaling law yield comparable performance to WSD in context of Figure 2.

---

> ### Author Response · Authors · 2024-11-24
> **Thanks a lot for your further feedback!**
>
> Regarding the corner case of appending zero learning rates, we suggest researchers to change the scaling law form to $S_2=\sum \limits_{i=1}^{s}m_i\cdot \eta_{i}^{\epsilon}$ (re-fitting and re-predicting) when appending zero learning rates, which can well avoid the undesired behavior you mentioned. May you specify the reason why you think the form $S_2=\sum \limits_{i=1}^{s}m_i\cdot \eta_{i}^{\epsilon}$ is not a solution to the issue?
>
> Regarding the issue for the optimal learning rate schedule, it is not an easy thing to solve the **exact** optimal LR schedule, while we have presented a nearly optimal LR schedule (WSD + 10-20% annealing ratio + 1-sqrt annealing function) in this paper. We elaborate this issue from two aspects:
>
> 1. **From a practical engineering aspect**, we have indeed predicted the (nearly) optimal learning rate schedule in this work. Specifically, WSD LRS should be used to confirm the larger values for both $S_1$ and $S_2$, as stated in section 4.2 and Appendix G.2; WSD LRS annealing ratio can be determined by the method stated in section 4.3; Annealing function can be determined by the method stated in Appendix G.3. In practical engineering, it is quite easy to get the (nearly) optimal learning rate schedule based on the composition of the methods above. In Fig. 24, we compare the prediction and our empirical results under a better annealing function and annealing ratio, where the prediction and ground-truth curves coincide very well. **Notably, we exactly use the equation as in Fig. 2, $L(s)=2.628+0.429⋅S_1^{−0.550}−0.411·S_2$,  to predict and draw figures in all parts of section 4 and Appendix G**.
>
> 2. **From a mathematical aspect**, it becomes an optimization problem, that is, searching a learning rate sequence $(\eta_1, \eta_2, \cdots, \eta_s)$, to optimize $\text{max }L(s)= L_0 + A\cdot S_1^{-\alpha} - C\cdot S_2$ .  Given the large search space of for solving the **high dimensional non-linear** problem, this might be not that easy as you think.  We can not find an exact optimal LR schedule solution based on mathematical deductions from our previous attempts. We would like to conduct some experiments if you can solve it and specify an optimal learning rate schedule. From another perspective, we believe that the solution can be built up by some non-linear optimization algorithms (e.g. neural networks + gradient decent) to get a **locally** optimal solution, but the minimization is local and the found LR schedule would be very like the schedule we mentioned above (i.e., WSD + 10-20% annealing ratio + 1-sqrt annealing function). Overall, this is a quite interesting thing on which the following work can have deeper explorations. At this stage, we believe that the approach based on the practical engineering mentioned above is sufficient to obtain a (nearly) optimal learning rate schedule, and the exprimental results coincide well with the predictions.
>
> We look forward to further discussion if you have any more questions and concerns.

---

> > ### Comment · Reviewer_up2N · 2024-11-24
> > **Exact Solution of Optimal LR schedule**
> >
> > If I understand correctly, we need to solve the following constrained convex optimization problem (I changed the symbol for the number of the total steps from $s$ to $T$ for my convenience):
> >
> > $$\min_{\eta_{1:T}\in[0,\eta_0]^T} L(\eta_{1:T}) = L_0 + A\cdot S_1^{-\alpha} - C\cdot S_2$$, where $S_1\triangleq \sum_{t=1}^T\eta_t$, $S_2 = \frac{1-\lambda^T}{1-\lambda}\eta_0 - \sum_{t=1}^T\lambda^{T-t} \eta_t.$
> >
> > First this problem is convex when $\alpha>0$, $A>0$. Therefore therefore a solution is a minimizer if and only if it is a KKT point. The minimizers exist because the objective is continuous and the domain is closed and bounded.
> >
> > For a KKT point $\eta_{1:T}$, there are only three possibilities for each $\eta_t$: (1). $\eta_t = \eta_0$, $\partial L/\partial \eta_t\le 0$; (2)  $\eta_t = 0$, $\partial L/\partial \eta_t \ge 0$; (3). $\partial L/\partial \eta_t = 0$.
> >
> > Note that the partial derivative of L with respect to $\eta_t$ becomes:
> >
> > $$\frac{\partial L}{\partial \eta_t} = -\alpha A S_1^{-\alpha-1} \frac{\partial S_1}{\partial \eta_t} + C \frac{\partial S_2}{\partial \eta_t} = -\alpha A S_1^{-\alpha-1} + C \cdot (-\lambda^{T-t}),$$
> >
> > there will be at most one $t\in[1,T]$ such that $\partial L/\partial \eta_t = 0$. That means the optimal LR schedule $\eta_{1:T}$ satisfies that exists some $0\le t^*\le T$, such that $\eta_t = \eta_0$ for all $t\le t^*$, $\eta_t = 0$ for all $t\ge t^*+2$. This implies that $$0\ge \frac{\partial L}{\partial \eta_{t^*}} = -\alpha A S_1^{-\alpha-1} + C \cdot (-\lambda^{T-t^*}) = -\alpha A (t^*\eta_0)^{-\alpha-1} + C \cdot (-\lambda^{T-t^*}),$$ and that $$0\le \frac{\partial L}{\partial \eta_{t^*+1}} =  -\alpha A S_1^{-\alpha-1} + C \cdot (-\lambda^{T-t^*-1}) \le -\alpha A ((t^*+1)\eta_0)^{-\alpha-1} + C \cdot (-\lambda^{T-t^*-1}).$$
> >
> > $t^*$ can be solved from the above two inequalities via a binary search for roots of $f(t^*) = -\alpha A (t^*\eta_0)^{-\alpha-1} + C \cdot (-\lambda^{T-t^*}) =0$. Once we know $t^*$, $\eta_{t^*+1}$ can be solved by checking either possibility (2) or (3) happens for $\eta_{t^*+1}$.
> >
> > If the above analysis is correct, having zero learning learning rates in the end is not a corner case for the proposed scaling law, but likely the optimal case. I look forward to seeing the experimental result on this optimal schedule.

---

> ### Author Response · Authors · 2024-11-25
> **Thank you for the detailed analysis and more clarifications**
>
> Thank you for the detailed analysis for our main format in eq.1.
>
> First, we apologize for the possible confusion brought by the phase "corner case" in our response. We intend to underline the fact that few LLM participants will pad 0 LR steps at the end of their training. We start our study with the intention of covering the most common use cases. Our extensive experiments (acknowledged by Reviewer `rnjo`, `up2N`) have demonstrated the effectiveness of our equation in these most commonly used learning rate schedulers, such as WSD, cosine, constant, multi-step cosine, and cyclic scheduler (acknowledged by Reviewer `rnjo`, `up2N`, and `5e85`).
>
> We would like to point out that your math deduction has a set of typos. Specifically, it should be ${\partial{L}}/{\partial{\eta_t}}=-\alpha A S_1^{-\alpha-1} {\color{red}{-}} C \cdot (-\lambda^{T-t})$ for the partial derivative of $L$ (*our simple main format*). Besides this minor issue, your analysis (thanks a lot!) is correct: the solved optimal learning rate schedule has zero learning rates in the end.
>
> As we have discussed in our paper (as well as in our responses to all reviewers), this problem is related to the issue ($\eta \approx 0$ case) that mentioned in Appendix H.3. Specifically, our main format does not well handle the case $\eta=0$ at last and the optimization leads to the unhandled case. Mathematical optimization strongly depends on an absolute accuracy, while our scaling law in such scenario does not achieve perfectly 100% accuracy (shown in Fig. 3). In comparison, **our mentioned variant forms** (e.g., Eq. 10, $S_2=\sum \limits_{i=1}^{s}m_i\cdot \eta_{i}^{\epsilon}$) with extra parameters should be more preferably used to solve the optimal LRS, because they cover more corner cases and have higher accuracy. It won't lead to the same optimal learning rate schedule as our main format if solving it using your mentioned method.
> We also believe that, in the future, there will be stronger and more parameterized specific forms, which are more suitable for directly mathematically solving the optimal LR schedule. At this stage, we believe that the approach based on the practical engineering elaborated in this paper is sufficient and more suitable to obtain a (nearly) optimal LR schedule.
>
> We would like to emphasize our point of view about this issue (as we also stated in global response before):
>
> - Studies of scaling laws are mostly empirically driven. Over-parameterizing the scaling law equation with more parameters essentially leads to more accurate fitting results. However, it will also complicate the final format and hinders us to focus on major factors for the training dynamics. For example, we compared our main format ( $L \propto S_2$) with $L \propto S_2^{\zeta}$ (adding a parameter $\zeta$ as power of $S_2$) in Fig. 29 of Appendix H.3, and the latter one is indeed better (marginally). **We choose our main format not due to absolute prediction accuracy but to pursue the simplification (i.e., fewest extra parameters) to model the essential training dynamics of LLMs.** Notably, in our main format in Eq.1, we only introduce one extra parameter compared to Chinchilla scaling law, i.e., the coefficient $C$ of the $S_2$ term. Moreover, it has been acknowledged by all the reviewers that our main scaling law format has a strong and robust capacity across many practical scenarios. We even validate our scaling law under random learning rate schedule in the response to another reviewer `rnjo`.
>
> - We agree with you and another reviewer `5x7q` that the current format might not be perfect. However, we want to highlight that our novel formulation (acknowledged by all reviewers) is the first to address the learning rate scheduling process in scaling law studies (acknowledged by reviewer `5x7q`), and we believe and expect that there should be more powerful specific format (maybe with more parameters) after this work.

---

> ### Author Response · Authors · 2024-11-27
> **Revisions Based on Your Feedback & Looking Forward to Your further Reply**
>
> Dear Reviewer up2N,
>
> Thank you once again for your thoughtful review and detailed analysis.  Based on your valuable feedback, we have made several revisions to the paper.
>
> Specifically, we have elaborated further on the reasons for selecting our main format in Appendix H.3, emphasizing its simplicity and effectiveness in most practical scenarios. Additionally, we have added a new subsection (Appendix H.4) to the Discussion section to discuss the issue regarding the optimal LR schedule that you raised. Here is the last paragraph in Appendix H.4:
>
> > From another perspective, one might adopt our scaling law to directly solve optimal LR schedule mathematically. It could be found that our Eq.1 leads to a collapsed LRS: some zero learning rates at last. This problem is related to the issue ($\eta \approx 0$ case) that we discussed above in Appendix H.3.
> Mathematical optimization strongly depends on an absolute accuracy, while our scaling law in such scenario does not achieve perfectly 100\% accuracy (shown in Fig. 3).
> In comparison, our mentioned variant forms (e.g., Eq. 10) with extra parameters should be more preferably used to solve the optimal LRS, because they cover more corner cases and have higher accuracy. We believe that in the future, there will be stronger and more parameterized specific forms, which are more suitable for directly mathematically solving the optimal LR schedule. At this stage, we believe that the approach based on the practical engineering mentioned above is sufficient to obtain a (nearly) optimal LR schedule.
>
> We hope these revisions adequately address your concerns, and we look forward to hearing your further thoughts on our updates. We are truly grateful for your insightful comments and remain committed to improving our work based on your suggestions.
>
> Best regards,
>
> Authors

---

> > ### Comment · Reviewer_up2N · 2024-12-03
> >
> > I appreciate the authors' clarification. However, this does not resolve my concern. If the other format of scaling law predicts an optimal schedule that works well in practice, the authors should present the theoretical and empirical evidence for that, instead of  just touching it briefly in appendix as a future direction. I think the ability of generalizing beyond training data is much more important than simplicity for a scaling law.
> >
> > This is still a very interesting work though. I am convinced by the formula of S_1, but still skeptical about S_2. The fact that we cannot optimize using it suggests the current formula of S_2 doesn't really capture the mechanism of LR decay. I would like l remain my current score.

---

> ### Author Response · Authors · 2024-12-03
> **Serious Misunderstandings & Clarifications**
>
> 1. The goal of this work is NOT to solve mathematically optimal LR schedule though we discuss much about this issue in rebuttal. Here is part of our abstract to present our motivations and contributions:
>
>    > This approach significantly reduces computational cost in formulating scaling laws while providing more accuracy and expressiveness. Extensive experiments demonstrate that our findings hold across a range of hyper-parameters and model architectures and can extend to scaling effect of model sizes. Moreover, our formulation provides accurate theoretical insights into empirical results observed in numerous previous studies, particularly those focusing on LR schedule and annealing.
>
>
> 2. **Our scaling law does have great generalization**. In Fig. 3, these predicted full loss curves are all under **unseen** LR schedules including WSD, cyclic, etc. The training data (in Fig. 2) only includes constant and cosine LR schedules. The generalization ability of our scaling law is also acknowledged by reviewer `5e85`, `5x7q`.
>
>    For those really uncommon LR schedules (e.g., padding zero LR at last), we recommend to use our variant form, which is explicitly mentioned in Line 286 and Line 527-528. The variant forms share the same core idea including $S_1$ and the momentum form in $S_2$, which we have demonstrated in Sec. 3.1 and Sec. 3.2. The variant forms are more accurate in uncommon LR schedules but introduce one extra parameter, for which we use the term "simplicity" to describe our main form.

---

### Official Review · Reviewer_rnjo · 2024-11-08

**Soundness:** 3
**Presentation:** 2
**Contribution:** 3
**Rating:** 6
**Confidence:** 4

**Summary:**

This paper proposes a scaling law that predicts the full loss curve of LLM pretraining: $L(s) = L_0 + A \cdot S_1^{-\alpha} - C \cdot S_2$. Extensive experiments demonstrate the law's effectiveness across many LLM pretraining settings. It can also be used to guide practitioners in choosing hyperparameters when setting the LR schedules.

**Strengths:**

1. Deriving a more accurate scaling law for LLM pretraining is an important topic. This paper presents a novel scaling law for loss curve prediction.
2. The proposed scaling law is validated in many LLM pretraining settings.
3. The proposed scaling law can be applied to guide practitioners in choosing hyperparameters when setting the LR schedules.

**Weaknesses:**

1. It is not very clear what kind of LR schedules this scaling law is applicable to and how large errors we will get for different schedules. I understand the authors' point that the law can be applied to various practical learning rate schedules. Still, it is easy to find or imagine many failure cases.
    * If we use an extremely large learning rate (say, 100) once in a while, the pretraining loss will go up to the level of random guessing repeatedly. Near the end of training, we turn to use very small learning rates. The proposed scaling law in the paper would suggest that doing this will lead to a much smaller loss than training without the extremely large learning rates, but it is hard to believe it to be true in practice.
    * This law is invariant to padding zero learning rates at the end. For a training run with some learning rate schedule, adding more steps with zero learning rate should not change the final training loss. However, if we pad zeros at the end of the learning rate, the scaling law will predict a much smaller loss as $S_2$ will gradually increase to its maximum possible value $\eta_0$. There is a related discussion on this in Appendix H.3, and the authors proposed to add a multiplicative factor at some places of the formula. But this does not really lead to a different loss formula because the factor $\approx 1$ after fitting real data (i.e., $\epsilon \approx 0, \eta_i^\epsilon \approx 1$).
    * For the learning rate schedules in Figures 3(c) and 3(d), the scaling law's predictions deviate from the actual values by quite a lot.
    * If at each step, I randomly choose the current learning rate from a uniform distribution (e.g., U[0, 1e-4]), will the scaling law still work?
2. While the title of Section 3 is called “Theory,” there is no real theoretical analysis in the paper to justify the proposed scaling law. The power law form for $S_1$ is obtained through speculation. The momentum form of $S_2$ is obtained from the observation of delay phenomenon, but it doesn't exclude the possibility that other forms could be better, e.g., sliding window average, or the same momentum form but the momentum coefficient $\lambda$ depends on learning rates. I acknowledge that proving any theorem about the loss curve in deep learning is hard, but I would like to see more evidence that every component of the scaling law is important (and perhaps irreplaceable). Doing some ablation studies could also be informative.

**Questions:**

It is an interesting paper overall, but I would like to see the authors' response to the two weaknesses I listed above.

=============

Post-rebuttal update:

I'm satisfied with the authors' response and would like to thank them for the additional experiments and clarification.

Assuming that the authors will further clarify to what schedules their scaling laws are applicable, I'm willing to raise my score to 7.
* Finding a good scaling law to capture LR schedules is challenging, and I don't think we should reject a paper because it has some failure cases. But the paper has to make enough efforts to search for failure cases and report them clearly for future exploration.
* I would encourage the authors to think more deeply about the precise definitions of "common" and "corner" cases. What properties are shared by widely adopted schedules that make the proposed scaling law work in practice?

As only 6 and 8 options are available, I have to keep my score 6 in the system.

---

> ### Author Response · Authors · 2024-11-16
> **Response to Reviewer rnjo [1/4]**
>
> ## Q1: An extremely large learning rate induces better result in the scaling law.
>
> Thanks for your valuable feedbacks and there might be some misunderstandings about our scaling law.
>
> 1. We present the restrictions for our scaling law in Line 279-282:
>
>    > Given the same training and validation dataset, the same model size, the same training hyper-parameters such as warmup steps, **max learning rate $η_{max}$** and batch size, the language modeling loss at training step $s$ empirically follows the equation $L(s) = L_0 + A\cdot S_1^{-\alpha}-C\cdot S_2$, where S1 and S2 are defined in Eq. 1. $L_0$, $A$, $C$, $\alpha$ are positive constants.
>
>    **Our scaling law cannot predict across different max learning rates using one single set of parameter.** Loss curves with different max learning rates have totally different $L_0$, $A$, $C$, $\alpha$. When setting a larger learning rate, the max learning rate also becomes larger and the orginal parameters, $L_0$, $A$, $C$, $\alpha$ are not suitable anymore. It is very likely that the newly fitted parameters under the condition of larger $\eta_{max}$, will lead to a worse prediction result compared with the condition of smaller $\eta_{max}$.
>
>    - For example, in our main experiments (Fig. 2 and Fig. 3),  we use the equation $L(s) = 2.628 + 0.429 \cdot S_1^{−0.550} - 0.411 \cdot S_2 $ for $\eta_{max}=2e-4$. We control other all conditions and only increase $\eta_{max}$ to $4e-4$ and re-train the models, then we get a different equation after fitting: $L(s) = 2.680 + 0.491\cdot S_1^{-0.639} - 0.253\cdot S_2$  ($R^2>0.999$).
>
>    Therefore, our scaling law **never** encourages to make learning rate as large as possible (e.g. 100). Conversely, When we set another max learning rate, the scaling law should be re-fitted and we get another set of parameters. Thus, we cannot directly compare the losses between larger and smaller learning rates only through the scaling law formulation. Instead, we should substitute specific parameters into the different equations for comparison between different $\eta_{max}$.
>
>
>
> 2. Our scaling law is used to fit and predict the loss curves when the curve is **not divergent**. It does not even follow a power law (Chinchilla and Openai scaling law) if the curve is divergent. Infinitely large LR would indeed "go up to the level of random guessing repeatedly", and inevitably lead to a divergent loss curve, which is out of scope of this work.
>
> Thanks for your valuable feedbacks again! We will also emphasize this retrictions in our revised manuscript.
>
>
>
> ## Q2: This law is invariant to padding zero learning rates at the end.
>
> We are glad that you have noticed this issue and found that we have discussed this issue in Appendix H.3. Here we give some further clarifications for this issue.
>
> 1. We designed a variant of our scaling law by adding a LR-weighted coefficient to $S_2$, where $S_2=\sum \limits_{i=1}^{s}m_i\cdot \color{red} \eta_{i}^{\epsilon}$ and $\epsilon$ is a newly added parameter. This format can well address the issue that “adding more steps with zero learning rate always reduces the predicted final loss”, because it makes the new terms of $S_2$ become zero when learning rate is $0$. We also footnoted in the paper (in Appendix H.3) that we recommend future researchers to try this format for fitting if possible.
>
> 2. For clafication, **we do not deny the variant** $S_2=\sum \limits_{i=1}^{s}m_i\cdot \eta_{i}^{\epsilon}$. As you noticed, the fitting result $\epsilon \approx 0$ is **based on our real data** (i.e., cosine LRS + constant LRS as fitting data), which contains few learning rates that are extremely near $0$.  In most practical situations, the learning rate will not be (extremely near) zero, making this variant unnecessary for practical use. Conversely, when some extreme corner cases really happen (e.g., very small learning rates are padded), we recommend future researchers to try this variant format, and the results would be expected to get different and much better because $\epsilon$ is no more near $0$.
>
> 3. Moreover, we would like to present some more explanations about why we choose our main format $S_2=\sum \limits_{i=1}^{s}m_i$. Extensive experiments have validated the effectiveness of our equation across many LLM pretraining settings (acknowledged by reviewer `rnjo`, `up2N` and `5e85`). This demonstrates the robustness of our equation in practical applications. Moreover, we can also easily extend our formulation to model some extremely rare corner cases. We discuss and explain more about the form of $S_2$ in the following Q5, Q6, and Q7.

---

> ### Author Response · Authors · 2024-11-16
> **Response to Reviewer rnjo [2/4]**
>
> ## Q3: For the learning rate schedules in Figures 3(c) and 3(d), the scaling law's predictions deviate from the actual values by quite a lot.
>
> We state the issue about accuracy of our scaling law from the following three aspects:
>
> 1. We would like to point out that this prediction error is already very small. As shown in the **middle** sub-fig in Fig. 3(c) and 3(d), the predicted curve aligns very well with the ground truth. The seemingly large deviation in the zoomed-in view on the right is due to the vertical axis being scaled down to **a very small range of 0.01.** This indicates that the maximal absolute loss error in both cases is approximately only **0.01,** which significantly outperforms many related scaling law papers. Here we list some famous and recognized scaling law works for comparison:
>
>    - In OpenAI scaling law paper [1], in their Fig. 1 (medium), they use 0.3 as tick interval (30x of ours), and that is even a fitting result rather than prediction. Some points are obviously more deviated from the actual values than ours.
>
>    - In OpenAI scaling law paper [1], in their Fig. 4 (left), they use 0.5 as tick interval (50x of ours), and shows significent deviated prediction results.
>
>    - In GPT-4 paper [2], in their Fig. 1, they use 1.0 as tick interval (100x of ours) and get a less accurate prediction result than ours.
>
>    - In Chinchilla scaling law paper [3], in their Fig. 3 (left), they use 0.2 as tick interval (20x of ours), and some points are obviously more deviated from the actual values than ours.
>
>    - In Llama-3 paper [4], in their Fig. 2 (left), they use 0.05 as tick interval (5x of ours), and some data points are more deviated from the actual values than ours.
>
>    - In Llama-3 paper [4], in their Fig. 4 (left), they use 0.025 as tick interval (2.5x of ours), and the prediction result is worse than ours.
>
>    - In DeepSeek paper [5], in their Fig. 4(a), they use 0.2 as tick interval (20x of ours), and some points are obviously more deviated from the actual values than ours.
>
>    - In Baichuan-2 paper [6], in their Fig. 4, they use 2.0 as tick interval (200x of ours), and get a much less accurate prediction result than ours.
>
> ​Overall, we would like to emphasize that our scaling law is quite accrate compared with other scaling law works. We set **0.01** as our tick interval to provide readers with details of the predicted curves. We believe that it is a more decent practice than hiding the error using some presentation tricks.
>
> 2. We did not include any WSD learning rate curves or learning rate curves with re-warmup in the fitting process.  Instead, our current fitting data points are based solely on two different and separate schedulers (i.e. cosine and constant), and are limited to just 20K steps. Moreover, there could be more other factors influencing the validation loss. For example, some training samples near specific checkpoints are more similar to the validation set, and then it could make a temporarily lower validation loss. Overall, the task is quite difficult but our scaling law gives an accurate enough prediction.
>
> 3. Our scaling law equation can easily make more accurate predictions if more data points are used in fitting. For example, if we add loss curves obtained from WSD LRS (50K steps, 20% 1-sqrt annealing to 0), the equation becomes $L(s) = 2.611 + 0.449\cdot S_1^{-0.532} - 0.330 \cdot S_2$ ($R^2>0.999$). The new equation is more accurate in predicting Fig 3(c) and 3(d), and reducs the mean prediction error by almost 60%. Here we present a table below to show the prediction improvement by adding more fitting data.
>
>    | Fitting Points                   | Equation                                                    | Fig. 3(c) mean prediction error | Fig. 3(d) mean prediction error |
>    | -------------------------------- | ----------------------------------------------------------- | ------------------------------- | ------------------------------- |
>    | Cosine + Constant (in our paper) | $L(s) = 2.628 + 0.429 \cdot S_1^{−0.550} - 0.411 \cdot S_2$ | 0.235%                          | 0.322%                          |
>    | Cosine + Constant + WSD          | $L(s) = 2.611 + 0.449\cdot S_1^{-0.532} - 0.330 \cdot S_2$  | 0.095%                          | 0.273%                          |
>
>
>
> [1] [Scaling Laws for Neural Language Models](https://arxiv.org/pdf/2001.08361)
>
> [2] [GPT-4 Technical Report](https://arxiv.org/pdf/2303.08774)
>
> [3] [Training Compute-Optimal Large Language Models](https://arxiv.org/pdf/2203.15556)
>
> [4] [The Llama 3 Herd of Models](https://arxiv.org/pdf/2407.21783)
>
> [5] [DeepSeek LLM: Scaling Open-Source Language Models with Longtermism](https://arxiv.org/pdf/2401.02954)
>
> [6] [Baichuan 2: Open Large-scale Language Models](https://arxiv.org/pdf/2309.10305)

---

> ### Author Response · Authors · 2024-11-16
> **Response to Reviewer rnjo [3/4]**
>
> ## Q4: If at each step, I randomly choose the current learning rate from a uniform distribution (e.g., U[0, 1e-4]), will the scaling law still work?
>
> This is a quite interesting experimental design! We conduct this experiment!
>
> The conclusion is: **our scaling law still works well, though the prediction error might be a bit larger.**
>
> For replication, we present the detailed experimental processes:
>
> 1. We first generate a random learning rate sequence using  `np.random.uniform(0, 2e-4, 30000)` after we set the seed by `np.random.seed(1442)`  (Numpy version: 1.24.4). The generated learning rate sequence is: `[6.82985668e-05, 1.81342066e-04, 7.20995317e-05 ... ]`. Please feel free to replicate and check the process above.
>
> 2. Then, we adopt setting B (shown in Table 3 and Appendix D.1, i.e. 293M model) for faster training, and the equation is ready-made: $L(s) = 2.761 + 0.517 \cdot S_1^{−0.491} - 0.458 \cdot S_2$. We launch a training for 30K steps according to the above random learning rate schedule after we warmup learning (100 steps) rate from 0 to 2e-4. Here we set U[0, 2e-4] because the max learnining rate of setting B is 2e-4.
>
> 3. Here is the prediction and ground-truth loss comparison table:
>
> | Steps (K)    | 3     | 6     | 9     | 12    | 15    | 18    | 21    | 24    | 27    | 30    |
> | ------------ | ----- | ----- | ----- | ----- | ----- | ----- | ----- | ----- | ----- | ----- |
> | ground-truth | 3.570 | 3.363 | 3.247 | 3.194 | 3.141 | 3.128 | 3.089 | 3.073 | 3.046 | 3.036 |
> | Prediction   | 3.630 | 3.371 | 3.254 | 3.184 | 3.135 | 3.100 | 3.071 | 3.050 | 3.031 | 3.015 |
>
> 4. Finally, as the table above shows, the prediction results are really accurate, considering the task is quite challenging. We get a **0.605%** mean prediction error, which is about 3x error compared to WSD LRS (Fig. 3c), and about 2x error compared to the Cyclic LRS (Fig. 3d). Notably,  as we mentioned above, this mean prediction error can be also further reduced by adding more fitting data.
>
> Interestingly, this random learning rate experiment can even serve as one another evidence for the power-law form of $S_1$, which is related to the following question you asked (Q5). We notice that in random learning rate schedule, the $S_2$ is quite stable over the training steps, while what really impacts the loss is $S_1$. The values of $S_1$ and $S_2$ are as follows (actually this can be easily replicated using several lines code based on the learning rate sequence generator above):
>
> | Steps (K) | 3     | 6     | 9     | 12    | 15    | 18    | 21    | 24    | 27    | 30    |
> | --------- | ----- | ----- | ----- | ----- | ----- | ----- | ----- | ----- | ----- | ----- |
> | $S_1$     | 0.315 | 0.617 | 0.919 | 1.222 | 1.524 | 1.827 | 2.124 | 2.423 | 2.723 | 3.020 |
> | $S_2$     | 0.094 | 0.099 | 0.100 | 0.099 | 0.100 | 0.099 | 0.102 | 0.101 | 0.100 | 0.100 |
>
> It is $S_1$ that is significantly different between random learning rate and constant learning rate ($S_1 = \eta_{max}\cdot s$ in constant LRS), and then finally makes a difference to the losses. We believe this supports our idea of using the powers of $S_1$ to predict the loss.
>
> Thanks for designing such a convincing and interesting experiments, which helps to comprehensively show the performance of our scaling law.
>
>
>
> ## Q5: Theoretical analysis of $S_1$ and $S_2$
>
> We are very glad that you are interested in the more theoretical justifications of our scaling law. We honestly admit that the form of $S_1$ and $S_2$ are primarily based on our empirical observations (i.e. Section 3.1 and 3.2). And in Q4 above, we also use your designed experiments (random LRS) to validate the form of $S_1$.  We have even claimed that our scaling law is empirical in the first line of the abstract.
>
> However, we would like to point out that even the Chinchilla and OpenAI scaling laws are, in fact, empirical, as explicitly stated in their respective papers. As you said, proving any theorem about the loss curve in deep learning is hard, and we believe that in AI domain, theoretical proofs often follow empirical works. For example, [7] presents some reliable theoretical justifications for scaling law after three years since OpenAI scaling law paper is published. We are continuously working on the theoretical aspects and this is indeed a very important future work.
>
> Actually, surprisingly, some related works have emerged very recently [8]. Their theoretical derivations are well matched with our scaling law. We would greatly appreciate it if you could kindly take the time to read it.
>
> [7] [The Quantization Model of Neural Scaling](https://arxiv.org/pdf/2303.13506)
>
> [8] [Understanding Warmup-Stable-Decay Learning Rates: A River Valley Loss Landscape Perspective](https://arxiv.org/pdf/2410.05192)

---

> ### Author Response · Authors · 2024-11-16
> **Response to Reviewer rnjo [4/4]**
>
> ## Q6: Some other possible $S_2$ form.
>
> Thanks for your great and thoughtful specific form suggestions! We have the following clarifications:
>
> 1. For the sliding window form, it is actually a "hard" version of our momentum form. Sliding window form assigns the same weight to each one in the window and drops ones out of the window. Our momentum form assigns weight based on the historical distance. They are consistent in their underlying ideas.
>
> 2. Essentially, over-parameterizing the scaling law equation (like parameterizing $\lambda$ as a function of $\eta$ ) with more parameters indeed often leads to more accurate fitting results. However, it will also complicate the final format and hinders us to focuse on major factors for the training dynamics. For example, we compared our main format ($L \propto S_2$) with $L\propto S_2^{\zeta}$ (adding a parameter $\zeta$ as power of $S_2$) in Fig. 29 of Appendix H.3, and the latter one is indeed better (marginally).
> **We choose our main format not due to absolute prediction accuracy but to pursue the simplification (i.e., fewest extra parameters) to model the essential training dynamics of LLMs**.
> Notably, in our main format in Eq.1, we only introduce one extra paramter compared to Chinchilla scaling law, i.e., the coefficent $C$ of the $S_2$ term but extend to the full loss curve prediction. Moreover, it has been acknowledged by all the reviewers that our main scaling law format has a strong and robust capacity across many practical scenarios. We also validate our scaling law even under random learning rate schedule in the above response.
>
> 3. We have elaborated further on the reasons for selecting our main format in Appendix H.3, emphasizing its simplicity and effectiveness in most practical scenarios. The revisions are highlighed in red.
>
>
>
> ## Q7: Ablation studies on $S_1$ and $S_2$
>
> Thanks for your valuable feedback! In the paper, we show the importance of $S_1$ by considering training disount in annealing (Line 212), and $S_2$ is necessary because there must exist some term to make the final loss decrease when LR annealing.
>
> We would like to also supplement some ablation studies on $S_1$ and $S_2$ to show that every component of the scaling law is important. Specifically, we compare some law forms without $S_1$ or $S_2$. We use the same setting as in Fig. 2 and Fig.3. For each format, we re-fit on 20K cosine + constant and re-predict on different LRS. The prediction results are shown as below:
>
> | Scaling law forms                                         | Prediction error on cosine LRS | Prediction error on multi-step cosine LRS | Prediction error on WSD LRS | Prediction error on Cyclic LRS |
> | --------------------------------------------------------- | ------------------------------ | ----------------------------------- | --------------------- | ------------------------- |
> | Ours: $L(s) = L_0 + A\cdot S_1^{-\alpha} - C\cdot S_2$    | 0.159%                         | 0.176%                              | 0.235%                | 0.322%                    |
> | w/o $S_1$: $L(s) = L_0 + A\cdot s^{-\alpha} - C\cdot S_2$ | 0.465%                         | 0.458%                              | 1.162%                | 1.139%                    |
> | w/o $S_2$: $L(s) = L_0 + A\cdot S_1^{-\alpha}$            | 1.261%                         | 1.265%                              | 1.519%                | 1.203%                    |
>
> The results suggest that the component of $S_1$ and $S_2$ is both important and Indispensable. We have also included the results in our revised paper (Table 4).
>
>
>
> Finally, we thank again for your valuable feedback! If your have any other questions or concerns, we are happy to answer and continually improve our work based on your feedback.

---

> ### Author Response · Authors · 2024-11-22
> **Please let us know if you have any further concerns.**
>
> We are very grateful for the opportunity to improve our work through your valuable feedback. If you think our response has addressed your concerns and questions, we sincerely hope you might consider raising your score. Should you have any further inquiries or require additional clarification, we warmly welcome your questions and are eager to provide comprehensive responses.

---

> ### Author Response · Authors · 2024-12-02
> **Kind reminder for your feedback**
>
> Dear Reviewer `rnjo`,
>
> We are very grateful for the opportunity to improve our work through your valuable feedback. As the discussion period is near the end, this is a kind reminder for your further feedback.
>
> If you think our response has addressed your concerns and questions, we sincerely hope you might consider raising your score. Should you have any further inquiries or require additional clarification, we warmly welcome your questions and are eager to provide comprehensive responses.
>
> Best regards,
>
> Authors

---

> > ### Comment · Reviewer_rnjo · 2024-12-02
> >
> > I would like to thank the authors for their long response. While I deeply appreciate the additional experiments, I still have the following concerns:
> >
> > 1. As also noted by Reviewer up2N, the current scaling law cannot work well with schedules with many zeros at the end. This is NOT a corner case because it is an optimal schedule predicted by the law. I understand that the authors have proposed some alternatives, but the scaling law in the main text is what the authors recommend the users to try.
> > 2. Still, as I mentioned in the review, it is not very clear what kind of LR schedules this scaling law is applicable to. According to the paper and the rebuttal, the authors' claim seems to be that the scaling law can work as long as the max learning rate is fixed. This is not true because the law is not invariant to padding zeros. Then, for what kind of schedules can the law work well? In the review, I only proposed 4 sanity checks, but I could list even more. E.g., what happens if the last LR jumps to the max LR? I don't mean the authors should endlessly add more experiments. Instead, I would like to suggest the following: if the authors want to claim that the scaling law can work except for “extremely rare corner cases,” then it would be better to properly define what cases are common cases and what cases are corner cases. Otherwise, I could argue that ending the schedule with many zeros is not a corner case at all (as discussed above), and I'm sure it is not the only counterexample. Adding some clarifications to narrow down the types of schedules that the law is applicable to would make the paper more rigorous and more scientific.
> > 3. (Minor Concern) Regarding Q3, the authors argue that their plots didn't look well because they used a more narrow tick interval. However, my concern lies not in the absolute errors but in the qualitative predictions. For example, it can be seen from 3(c) and 3(d) that the predicted curves do not always follow the same *trend* as the actual curve. This is a worrying behavior because it can lead to wrong predictions about what schedules could be better.
> >
> > Overall, I feel it is a good paper, and I would recommend acceptance. I will keep my score 6 due to the above issues of the paper.

---

> ### Author Response · Authors · 2024-12-02
> **Thanks for your further feedback!**
>
> We are sincerely grateful for your recommendation to accept this work! In response to your concerns, we have the following answers.
>
> **Regarding your Q1:** We have highlighted the following things updated in the latest version of our paper.
>
> - We further emphasize the restrictions of our scaling law in Line 283-285. And we mention our proposed variants to counter the case of zeros learning rates at last **in the main text** including Line 286 and Line 527-528.
> - We discuss why we choose our main format rather than our proposed two variant formats in Appendix H.3. Specifically, we prefer the simplification (i.e., the fewest parameters) to model the essential dynamics in LLM training.
> - We add one subsection to explicitly discuss on solving optimal learning rate schedule in Appendix H.4.
>
>
>
> **Regarding your Q2:** Thanks for your suggestions. We would like to clarify the point about common and corner LR schedules.
>
> - Common LR schedules mean popular LR schedules that many previous researchers widely adopt, including constant, cosine, multi-step cosine, WSD, and cyclic LR schedule. These schedules are exactly the cases shown in Fig. 2 and Fig. 3.
>
> - The "corner" LR schedules mean those LR schedules that few previous researchers adopt, including the case with many zero LR at last, and random learning rate schedule you mentioned (though we also validate our scaling law for random LR schedule in our previous discussion). We recommend our variant forms for these "corner" LR schedules if possible.
>
> We would like to further emphasize this point in the next version of the paper, as the deadline for revisions during the discussion period has passed.
>
>
> Thanks again for your recommendation for this paper. We are committed to continuously improving our work based on your valuable suggestions.

---

### Author Response · Authors · 2024-11-16
**Clarification for the restrictions of our scaling law [1/2]**

This is a response to all the reviewers. We are deeply grateful for the time and effort you have dedicated to the thorough review of our manuscript. Your insightful comments and suggestions are highly appreciated and have been extremely valuable in improving and enhancing the quality of this work.

We are gratified to see that our work has been acknowledged in many aspects including:
1. Our work is highly significant in the field of LLM pretraining (reviewers `rnjo`, `up2N`, and `5x7q`)
2. Our scaling law formulation has been recognized as innovative by reviewers `rnjo`, `up2N`, `5e85`, and `5x7q`.
3. The effectiveness of our scaling law across a variety of LLM pretraining settings has been acknowledged by reviewers `rnjo`, `up2N`, and `5e85`.

We noticed that the reviews `rnjo` and `up2N` have some common misunderstandings about the restrictions of our scaling law (we are pleased that the reviewer `5x7q` has noticed the restrictions we mentioned in our paper, and particularly pointed them out in the review.). We have the following clarifications on the restrictions of our scaling law.

## Q1:  larger (even infinite) learning rate induces lower loss.

There might be a misunderstanding about our scaling law. We present the restrictions for our scaling equation in Line 279-282:

> Given the same training and validation dataset, the same model size, the same training hyper-parameters such as warmup steps, **max learning rate $η_{max}$** and batch size, the language modeling loss at training step $s$ empirically follows the equation $L(s) = L_0 + A\cdot S_1^{-\alpha}-C\cdot S_2$, where S1 and S2 are defined in Eq. 1. $L_0$, $A$, $C$, $\alpha$ are positive constants.

Our scaling law cannot predict across different max learning rates using one single set of parameters. Loss curves with different max learning rate have different values of $L_0$, $A$, $C$, $\alpha$. When we set another max learning rate, the scaling law should be re-fitted and we get another set of parameters. Thus, we cannot predict losses for larger max learning rates with parameters fitted using a small max learning rates. Therefore, our scaling law **never** encourages to make learning rate as large as possible. Instead, we should substitute specific parameters into the different equations for comparison between different $\eta_{max}$.

For example, in our main experiments (Fig. 2 and Fig. 3),  we use the equation $L(s) = 2.628 + 0.429 \cdot S_1^{−0.550} - 0.411 \cdot S_2 $ for $\eta_{max}=2e-4$. We control all other conditions and only increase $\eta_{max}$ to $4e-4$ and re-train the models, then we get a different equation after fitting: $L(s) = 2.680 + 0.491\cdot S_1^{-0.639} - 0.253\cdot S_2$  ($R^2>0.999$).

Moreover, our scaling law is used to fit and predict the loss curves when the curve is **not divergent**. It is almost impossible to predict the behavior of a non-linear network if its loss diverge. Infinitely large LR would inevitably lead to a divergent loss curve, which is out of scope of this work. We will also emphasize this restrictions in our revised manuscript.

---

> ### Author Response · Authors · 2024-11-16
> **Clarification for the restrictions of our scaling law [2/2]**
>
> ## Q2: Adding more steps with zero learning rate always reduces the predicted final loss.
>
> We have discussed this scenario in "Section 6: Discussion" (Line 527-528) and introduced a possible variant of our equation in appendix H.3. Specifically, we can add a LR-weight coefficient to $S_2$, i.e., $S_2=\sum \limits_{i=1}^{s}m_i\cdot \color{red} \eta_{i}^{\epsilon}$ where $\epsilon$ is a new parameter. This format can well address the issue that “adding more steps with zero learning rate always reduces the predicted final loss”, because it makes $S_2$ becomes zero when the learning rate $\eta_{i}$ is $0$. We encourage other researchers to try this format (see the footnote in Appendix H.3).
>
> We choose to use $S_2=\sum \limits_{i=1}^{s}m_i$ as our main format for the following reasons (where we have briefly discussed in the Appendix H.3):
>
> 1. In our experiments, the absolute value of $\epsilon$ is extremely small after fitting. This makes the value of $\eta_{i}^{\epsilon}$ approaches to $1$ if the learning rate $\eta_{i}$ is within a normal scale, which is true in most practical training processes. The value of $\eta_{i}^{\epsilon}$ approaches to $0$ only when the learning rate $\eta_{i}$ is extremely small, which rarely happens in practical training.
>
> 2. Extensive experiments have validated the effectiveness of our equation across many LLM pretraining settings (acknowledged by reviewer `rnjo`, `up2N` and `5e85`). This demonstrates the robustness of our equation in practical applications. Moreover, we can also easily extend our formulation to model some extremely rare corner cases.
>
> 3. Essentially, over-parameterizing the scaling law equation with more parameters leads to more accurate fitting results. However, it will also complicate the final format and hinders us to focus on major factors for the training dynamics. Notably, in our main format in Eq.1, we only introduce one extra parameter compared to previous works, i.e., the coefficient $C$ of the $S_2$ term. We intend to use a concise scaling law format to reveal more insights of the training dynamic to LLM practitioners, and shed light on (1) studying the optimal learning rate schedule in pretraining; (2) predicting losses like Chinchilla scaling law but with much less training cost to collect fitting data.
>
> We agree with the reviewer `5x7q` that the current format might not be perfect. However, we want to highlight that our novel formulation (acknowledged by all reviewers) is the first to address the learning rate scheduling process in scaling law studies (acknowledged by reviewer `5x7q`).  The more deep investigation of extreme corner cases is left in further studies.

---

### Author Response · Authors · 2024-11-22
**Summary of the Revisions**

We sincerely thank the reviewers for their insightful comments on our paper. We have carefully considered all feedback and made corresponding revisions and improvements. Below is a summary of the main changes we have made to our manuscript.

1. We  add some descriptions to more emphasize the applicable scenarios of our scaling law in Line 284-286, based on the questions of reviewer `rnjo` and `up2N`.
2. We change the title of section 3 from "Theory" to "Observations and Experiments" to better reflect the nature of section 3, based on the review by reviewer `rnjo`.
3. We supplement ablation studies on $S_1$ and $S_2$ of our scaling law form in Table 4 of Appendix D.5, based on the suggestions of reviewer `rnjo` and `5x7q`.
4. We modify some sentences in section 4.4 and section 6 to better align with the suggestions by reviewer`5e85`.
5. We present more reasons for choosing our scaling law form in Appendix H.3, and supplement some discussions about solving exact optimal learning rate schedule using the scaling law in Appendix H.4, based on the suggestions of reviewer `up2N`.


All the revised sentences are highlighted in red. Besides, we provide detailed responses to each of the reviewers' comments as below.

Thank you once again for your review and suggestions. We believe these revisions have significantly enhanced the quality of the paper. We welcome any further feedback you may have.

---

### Meta-Review · Area_Chair_FTmt · 2024-12-23

**Metareview:**

This paper proposes a scaling law for predicting the full loss curve of language model training, accounting for both data size scaling and learning rate annealing effects. The authors claim their formulation can predict loss at any training step under various learning rate schedules using only one or two training curves for fitting, demonstrating results across different model architectures and hyperparameters.

The paper addresses an important aspect of LLM training by incorporating learning rate scheduling effects into scaling laws and shows potential for optimizing learning rate schedules. However, it suffers from significant weaknesses that ultimately lead to its rejection. The lack of robust theoretical justification for the proposed scaling law form is a major concern. Additionally, the paper fails to adequately address issues with corner cases, such as padding with zero learning rates, where the law breaks down. The inability to derive an optimal learning rate schedule from the proposed law further limits its practical utility.

The primary reason for rejection is the unresolved concerns raised by Reviewer up2N, particularly regarding the handling of corner cases and the lack of a clear optimal learning rate schedule derivation. The paper does not sufficiently explore the limitations of the proposed scaling law or provide a solid theoretical foundation. These issues indicate that the work requires substantial further development before it can be considered for publication.

**Additional Comments On Reviewer Discussion:**

The discussion centered on the paper's handling of corner cases, lack of theoretical justification, and inability to derive an optimal learning rate schedule. Reviewers raised concerns about the law's behavior in scenarios like padding with zero learning rates or using extremely large learning rates. They also questioned the lack of theoretical analysis supporting the proposed scaling law form and discussed whether the law could be used to derive an optimal learning rate schedule.

The authors provided responses and additional experiments to address these concerns, proposing variant forms for corner cases and acknowledging the empirical nature of their work. However, these responses did not fully resolve the fundamental issues raised by the reviewers, particularly those from Reviewer up2N. The authors suggested practical approaches for learning rate optimization but could not provide a definitive mathematical solution.

In weighing these points, the unresolved concerns from Reviewer up2N were the most significant factors in the decision to reject, as indicated by the Area Chair's comment. The paper's inability to address these core issues suggests that it requires substantial revision and development before it can be considered for publication.

---

### Decision · Program_Chairs · 2025-01-22

Reject